# The Potential Role of Cytotoxic Immune Effectors in Induction, Progression and Pathogenesis of Amyotrophic Lateral Sclerosis (ALS)

**DOI:** 10.3390/cells11213431

**Published:** 2022-10-31

**Authors:** Kawaljit Kaur, Po-Chun Chen, Meng-Wei Ko, Ao Mei, Nishant Chovatiya, Sara Huerta-Yepez, Weiming Ni, Sean Mackay, Jing Zhou, Dipanarine Maharaj, Subramaniam Malarkannan, Anahid Jewett

**Affiliations:** 1Division of Oral Biology and Medicine, The Jane and Jerry Weintraub Center for Reconstructive Biotechnology, University of California School of Dentistry, 10833 Le Conte Ave, Los Angeles, CA 90095, USA; 2Department of Microbiology and Immunology, Medical College of Wisconsin, Milwaukee, WI 53226, USA; 3IsoPlexis, 35 North East Industrial Road, Branford, CT 06405, USA; 4South Florida Bone Marrow Stem Cell Transplant Institute, DBA Maharaj Institute of Immune Regenerative Medicine, 10301 Hagen Ranch Rd Ste. 600, Boynton Beach, FL 33437, USA; 5Laboratory of Molecular Immunology and Immunotherapy, Blood Research Institute, Versiti, Milwaukee, WI 53226, USA; 6The Jonsson Comprehensive Cancer Center, UCLA School of Dentistry and Medicine, 10833 Le Conte Ave., Los Angeles, CA 90095, USA

**Keywords:** amyotrophic lateral sclerosis (ALS), NK cells, CD8+ T cells, IFN-γ, cytotoxicity, NAC

## Abstract

Amyotrophic lateral sclerosis (ALS) is an auto-immune neurodegenerative disorder affecting the motor-neuron system. The causes of ALS are heterogeneous, and are only partially understood. We studied different aspects of immune pathogenesis in ALS and found several basic mechanisms which are potentially involved in the disease. Our findings demonstrated that ALS patients’ peripheral blood contains higher proportions of NK and B cells in comparison to healthy individuals. Significantly increased IFN-γ secretion by anti-CD3/28 mAbs-treated peripheral blood mononuclear cells (PBMCs) were observed in ALS patients, suggesting that hyper-responsiveness of T cell compartment could be a potential mechanism for ALS progression. In addition, elevated granzyme B and perforin secretion at a single cell level, and increased cytotoxicity and secretion of IFN-γ by patients’ NK cells under specific treatment conditions were also observed. Increased IFN-γ secretion by ALS patients’ CD8+ T cells in the absence of IFN-γ receptor expression, and increased CD8+ T cell effector/memory phenotype as well as increased granzyme B at the single cell level points to the CD8+ T cells as potential cells in targeting motor neurons. Along with the hyper-responsiveness of cytotoxic immune cells, significantly higher levels of inflammatory cytokines including IFN-γ was observed in peripheral blood-derived serum of ALS patients. Supernatants obtained from ALS patients’ CD8+ T cells induced augmented cell death and differentiation of the epithelial cells. Weekly N-acetyl cysteine (NAC) infusion in patients decreased the levels of many inflammatory cytokines in peripheral blood of ALS patient except IFN-γ, TNF-α, IL-17a and GMCSF which remained elevated. Findings of this study indicated that CD8+ T cells and NK cells are likely culprits in targeting motor neurons and therefore, strategies should be designed to decrease their function, and eliminate the aggressive nature of these cells. Analysis of genetic mutations in ALS patient in comparison to identical twin revealed a number of differences and similarities which may be important in the pathogenesis of the disease.

## 1. Introduction

Amyotrophic lateral sclerosis (ALS), is a neurological syndrome initially recognized by Jean-Martin Charcot in 1869 as a pure motor neuron disease, characterized by progressive motor neuron loss in the brain and spinal cord [1,2,3]. ALS mostly has focal onset followed by spread to different body parts. Respiratory muscle failure limits the survival to 2–5 years after disease initiation [4]. The cause of ALS is not well-defined but appears to be heterogeneous. To date, over 20 genes have been found to be associated with ALS which indicates the complexity of the ALS genetics [2]. Family history can be found in 10% of ALS patients, but in 90% cases, no other family member is affected by ALS and is therefore classified as sporadic ALS [4,5]. Expansion in the C9orf72 gene was found in 30–50% of familial ALS and 7% of sporadic ALS [4]. Mutations of *superoxide dismutase 1, TDP-43,* and *FUS* have also been found in ALS [5,6,7]. To date, only supportive care is provided for the patients, and no effective treatment or cure has been found [2].

Recent studies indicated that the immune system plays an active role in ALS progression [8,9]. Neuroinflammation is the most prominent pathological finding associated with motor neuron injury characterized by monocyte and T cell infiltration, microglial activation, and astrogliosis [9,10,11]. CD4+ T cells were found to provide supportive neuroprotection in ALS patients by modulating the tropic/cytotoxic balance of glia [10]. Although more studies have been focused on CD4+ T cells from patients due to their protective role in the disease, the role of CD8+ T cells has received relatively less attention. CD8+ T cells were found in the spinal cord of ALS mice and ALS patients [12]. Activated CD8+ T lymphocytes infiltrating the central nervous system (CNS) of *(SOD1)^G93A^* mutant ALS mice were seen [12]. The reduced number of CD8+ T cells in these mice decreased spinal motor neuron loss. In addition, CD8+ T lymphocytes selectively killed motor neurons through peptide-MHC-I complex recognition.

Natural killer (NK) cells were found to be elevated in ALS patients’ peripheral blood [13] and could play a major role in ALS progression. NK cell-mediated cytotoxicity was also found to be elevated in ALS patients [14]. Although the studies in mice with one dominant mutation are very timely and important [15,16], they may not completely represent the human disease in which many gene mutations have been implicated. For these reasons, we studied the function of different immune subsets comprehensively in ALS patients to determine which main subsets could likely contribute to the disease induction and progression.

N-acetyl cysteine (NAC) inhibits cell death due to its anti-oxidant activity, in part through its ability to differentiate the stem cells [17,18,19]. It is also likely that NAC could protect differentiated cells from undergoing cell death. We have previously reported that NAC treatment differentiate stem cells of the apical papilla (SCAP), HEp2 oral epithelial cells and dental pulp stem cells (DPSCs) resulting in reduced functional loss and cell death of stem cells, and also increased *NF**κB* activity. *NF**κB* is a transcription factor responsible for DNA transcription, cytokine secretion and cell survival [20,21]. In the current study, ALS patients were given weekly NAC injections and the effect was studied on pro and anti-inflammatory cytokine release.

In addition, we comprehensively analyzed the phenotype of immune cell subsets in peripheral blood mononuclear cells (PBMCs) and determined the functions of PBMCs, NK cells, and CD8+ T cells of ALS patients in comparison to either their genetically identical twin or with other age and gender-matched healthy donors in order to determine whether correlations could be found with disease progression. We demonstrate the hyperresponsiveness of both NK and CD8+ T cells, and their significant increase in function under many therapeutic strategies targeting inflammation in ALS patients. Direct lysis by the cytotoxic cells, as well as over-differentiation and induction of cell death in motor neurons, due to the substantial increase in IFN-γ and TNF-α secretion by the CD8+ T cells and NK cells are likely underlying causes of motor neuron damage. In addition, we present the effect of NAC injection in patients on immune function. Our studies are significant since we compared the results of the identical twin or other healthy donors with the ALS patient immune function. More importantly, we followed the patient and his twin siblings’ peripheral blood immune function continuously for over four years to determine whether any changes or modifications in the immune parameters could be observed and correlated with the progression of the disease.

## 2. Materials and Methods

### 2.1. ALS Patients and Healthy Individuals’ Information

Healthy individuals were donors with no medical history of ALS disease and were the same age and gender as ALS patients. A total of 14 healthy individuals and 8 ALS patients were used in this study. We assessed the phenotype and functions of immune cells from genetically identical patient and healthy twin 25 times and patients with no genetic similarities 37 times over a period of two and half years.

#### 2.1.1. Cell Lines, Reagents, and Antibodies

RPMI 1640 (Life Technologies, Carlsbad, CA, USA) supplemented with 10% fetal bovine serum (FBS) (Gemini Bio-Products, West Sacramento, CA, USA) was used for peripheral blood mononuclear cells (PBMCs), NK cells, T cells, monocytes, and oral squamous carcinoma stem cells (OSCSCs) cultures. Recombinant IL-2 was obtained from NIH- BRB. Antibodies to CD16 (clone 3G8) were purchased from Biolegend, San Diego, CA, USA. ImmunoCult-XF T cell expansion medium and Immunocult human CD3/CD28 T cell activator was purchased from Stem Cell Technologies (Vancouver, BC, Canada). OSCSCs were isolated from patients with tongue tumors at UCLA [22,23,24,25]. Probiotic bacteria, sAJ2 is a combination of 8 different strains and is prepared as described previously [25], and RPMI 1640 supplemented with 10% FBS was used to re-suspend AJ2. Human ELISA kits for IFN-γ were purchased from Biolegend, San Diego, CA, USA. Chromium-51 radionucleotide was purchased from PerkinElmer, Richmond, CA, USA. The following antibodies with their respective clones were purchased from Biolegend, San Diego, CA, USA and used as suggested by the manufacturer for flow cytometric analysis; CD45 (H130) CD3/16/56: (UCHT1/3G8/MEM-188), CD3 (UCHT1), CD8 (HIT8a), CD14 (63D3) CD19 (HIB19).

#### 2.1.2. Isolation of Human PBMCs, NK Cells, T Cells, and Monocytes

Written informed consent approved by UCLA Institutional Review Board (IRB) was obtained from healthy individuals and ALS patients, and all procedures were approved by the UCLA-IRB. Patients were diagnosed as having ALS by their treating physicians, and genetic mutational analysis on the patient and his identical twin brother were performed in our collaborator’s laboratory Dr. Malarkannan. NAC treatments were received as part of the care from the patients’ attending physician. PBMCs were isolated from peripheral blood as described previously [26]. PBMCs were used to isolate NK cells, T cells, and monocytes using the EasySep^®^ Human NK cell, EasySep^®^ Human T cell, and EasySep^®^ Human monocytes enrichments kits, respectively, purchased from Stem Cell Technologies, Vancouver, BC, Canada.. Isolated NK cells, T cells, and monocytes were stained with anti-CD16, anti-CD3, and anti-CD14 antibodies, respectively, to measure the cell purity using flow cytometric analysis.

#### 2.1.3. Enzyme-Linked Immunosorbent Assays (ELISAs), Enzyme-Linked Immunospot (ELISpot), and Multiplex Cytokine Arrays

Single ELISAs and multiplex assays were performed as previously described [26]. To analyze and obtain the cytokine and chemokine concentrations, a standard curve was generated by either two- or three-fold dilution of recombinant cytokines provided by the manufacturer. The ELISpot was conducted according to the manufacturer’s instructions. The number of IFN-γ secreting cells was determined by using human IFN-γ single-color enzymatic ELISpot assay, and analyzed by the ImmunoSpot^®^ S6 UNIVERSAL analyzer and ImmunoSpot^®^ software (all CTL Europe GmbH, Bohn, Germany). For multiple cytokine arrays, the levels of cytokines and chemokines were also determined by multiplex cytokine arrays as recommended by the manufacturer (MAGPIX, Millipore, Billerica, MA, USA). Analysis was performed using a Luminex instrument (MAGPIX, Millipore, Billerica, MA, USA), and data were analyzed using the proprietary software (xPONENT 4.2, Millipore, Billerica, MA, USA). The range of sensitivity for detection of cytokine and chemokines in serum is from 0.4–55.8 pg/mL as reported by the manufacturer.

#### 2.1.4. Surface Staining

Staining was performed by labeling the cells with antibodies as described previously [26,27,28]. Flow cytometric analysis was performed using Attune^TM^ NxT Flow cytometer (Thermo Fisher Scientific, Waltham, MA, USA), and the results were analyzed by the FlowJo vX software (Ashland, OR, USA).

#### 2.1.5. ^51^Cr Release Cytotoxicity Assay

The ^51^Cr release assay was performed as described previously [29]. Briefly, 5:1, 2.5:1, 1.25:1, and 1:1 effector to target ratio was used to incubate effector cells with ^51^Cr–labeled target cells. After a 4 h incubation period, the supernatants were harvested from each sample and the released radioactivity was counted using the gamma counter. The percentage specific cytotoxicity was calculated as follows:

% Cytotoxicity = Experimental cpm − spontaneous cpm

Total cpm − spontaneous cpm

Lytic units (LU) 30/10^6^ is calculated by using the inverse of the number of effector cells needed to lyse 30% of tumor target cells ×100.

#### 2.1.6. NAC Preparation and Infusion

N-acetylcysteine (NAC) injections for intravenous solutions were prepared by adding varying doses of NAC powder (6000–8000 mg) to 500 mL of normal saline, after which the PH was measured and adjusted to 7. NAC infusion was delivered as part of patient care by the treating physician. Infusion of NAC was delivered at a slow rate on an outpatient basis. The patient gave informed consent for treatment using low-dose NAC IV infusions, including agreeing to the publication of the results.

#### 2.1.7. Single-Cell Protein Analysis and Polyfunctionality of NK and CD8+ T Cells

For on chip stimulation, NK cells were first labeled with membrane stain (1:500 dilution, IsoPlexis) and then resuspended in complete RPMI medium at a density of 1 × 10^6^ cells/mL with an addition of PMA (5 ng/mL; Sigma-Aldrich, P8139-1MG) and Ionomycin (500 ng/mL; Sigma-Aldrich, 10634-1MG), both PMA and Ionomycin were purchased from Millipore Sigma, Rockville, MD, USA. CD8+ T cells were treated with plate-bound anti-human CD3 (10 µg/mL; clone OKT3, Thermo Fisher/Invitrogen, Carlsbad, CA, USA) and soluble anti-human CD28 (5 µg/mL; clone CD28.2, Thermo Fisher/Invitrogen, Carlsbad, CA, USA) at a density of 1 × 10^6^ cells/mL for 24 h at 37 °C, 5% CO_2_. The stimulated CD8+ T cells were then labeled with membrane stain (1:500 dilution, IsoPlexis) for on chip cell detection and resuspended in complete RPMI medium at a density of 1 × 10^6^ cells/mL. Approximately 30 µL of CD8+ T cells or NK cells suspension (30,000 cells) was loaded into human adaptive IsoCode Chips (IsoPlexis) containing ~12,000 cellular microchambers, each of which was pre-patterned with a complete copy of the 32-plex antibody array for single-cell secretomic evaluation. Cells on the chip were incubated at 37 °C, 5% CO_2_ for additional 13.5 h on IsoLight automation system (IsoPlexis). Following this final incubation, subsequently secreted proteins from ~1000 single cells were captured by the 32-plex antibody barcoded chip and analyzed by backend fluorescence ELISA-based assay. Polyfunctionality of immune cells defined as a cell co-secreting 2+ cytokines were analyzed by the IsoSpeak software across the five functional groups: Effector (Granzyme B, TNF-α, IFN-γ, MIP1-α, Perforin, TNF-β); Stimulatory (GM-CSF, IL-2, IL-5, IL-7, IL-8, IL-9, IL-12, IL-15, IL-21); Chemoattractive (CCL11, IP-10, MIP-1β, RANTES); Regulatory (IL-4, IL-10, IL-13, IL-22, sCD137, sCD40L, TGF-β1); Inflammatory (IL-6, IL-17A, IL-17F, MCP-1, MCP-4, IL-1β). Protein signals from zero-cell microchambers were used to assess cytokine-specific background. Cutoffs for any given cytokine were computed based on background levels from wells not containing cells plus 3 standard deviations. In addition, signals with as signal-to-noise ratio (SNR) of at least 2 (relative to the background threshold) and from at least 20 single cells or 2% of all single cells (whichever quantity was larger) were considered as significantly secreted [30,31,32,33,34,35,36,37]. The functional groups of immune cells were deconvoluted and visualized by 3D t-Distributed Stochastic Neighbor Embedding (3D-tSNE) and heatmap visualizations. 3D t-SNE is a nonlinear dimensionality reduction tool used for visualizing multi-dimensional data in low-dimensional space (2D/3D) relying on computations based on algebraic topology and Riemannian geometry. Briefly, as the raw MFI (mean fluorescence intensity) data feeds into the t-SNE algorithm and is subsequently transformed/reduced, it calculates similarities between data points and then tries to optimize where the data point would end up in this 3D space. 3D t-SNE of all single cells was analyzed in the IsoSpeak software by using the following hyperparameters: theta: 0.5; perplexity: 50; maximum iterations: 1000.

#### 2.1.8. Sonication of Probiotic Bacteria AJ2

AJ2 is a combination of seven different strains of Gram-positive probiotic bacteria: *Streptococcus thermophiles, Bifidobacterium longum, Bifidobacterium breve, Bifidobacterium infantis, Lactobacillus acidophilus, Lactobacillus plantarum, and Lactobacillus casei*. AJ2 was weighed and re-suspended in RPMI 1640 medium containing 10% FBS at a concentration of 10 mg/mL. The bacteria were thoroughly vortexed and sonicated on the ice for 15 s at 6 to 8 amplitudes. Sonicated samples were then incubated for 30 s on ice, and the cycle was repeated for five rounds. After every five rounds of sonication, the samples were examined under the microscope until at least 80% of bacterial walls were lysed. It was determined that approximately 20 rounds of sonication/incubation on the ice were necessary to achieve complete sonication. Finally, the sonicated AJ2 (sAJ2) was aliquoted and stored at −80 °C until use.

#### 2.1.9. Generation of Osteoclasts and Osteoclasts-Induced NK Cell Expansion

Purified monocytes were cultured in alpha-MEM media supplemented with M-CSF (25 ng/mL) for 21 days and RANKL (25 ng/mL) for 21 days to generate osteoclasts (OCs). The media were replenished every three days. Human purified NK cells were activated with recombinant human IL-2 (rh-IL-2) (1000 U/mL) and anti-CD16 mAb (3 µg/mL) for 18–20 h before they were co-cultured with OCs and sAJ2 (OCs:NK:sAJ2; 1:2:4) in RPMI 1640 medium containing 10% FBS. The media were refreshed every three days with RPMI complete medium containing rh-IL-2 (1500 U/mL).

#### 2.1.10. Oral Squamous Carcinoma Stem Cells (OSCSCs) Treatment with CD8+ T Cells Supernatant

CD8+ T cells were treated with IL-2+anti-CD3/28 mAbs for 18–20 h before the supernatants were harvested to treat OSCSCs. For tumor cell death assay 3000–5000 pg IFN-γ containing supernatants was added for 4 days before tumors cell dead was determined with propidium iodine (PI) (100 μg/mL) staining using flow cytometric analysis. Differentiation of OSCSCs was conducted as described previously [23]. On average, a total of 3000–5000 pg of IFN-γ containing supernatants obtained from IL-2+anti-CD3/28 mAbs treated CD8+ T cells was added for 4 days to induce differentiation of OSCSCs.

#### 2.1.11. CD4+ T Cells Differentiation to Treg Cells

Naïve CD4+ T cells were isolated from PBMCs using negative isolation kit (cat # 19555) purchased from Stem Cell Technologies (Vancouver, BC, Canada). The isolated naïve CD4+ T cells were then differentiated into Treg using ImmunoCult™ Human Treg Differentiation Supplement Catalog # 10977 which is a Serum-free culture supplement for the differentiation of human naïve CD4+ T cells into regulatory T cells (Tregs) based on the manufacturer’s recommendation. Briefly, CD4+ T cells (1 × 10^6^ cell/mL) were cultured with ImmunoCult-XF T cell expansion medium and Immunocult Human CD3/CD28 T cell activator (25 µL/mL), both purchased from Stem Cell Technologies (Vancouver, BC, Canada). The cell density was adjusted to 1 × 10^6^ cell/mL every 3–4 days as needed with the addition of fresh medium. ImmunoCult™ Human Treg Differentiation Supplement contains a human cytokine and small molecule formulated to promote the robust activation, expansion, and differentiation of peripheral blood-derived, naïve, CD4+ human T cells into regulatory T cells (Tregs).

#### 2.1.12. Statistical Analysis

All statistical analyses were performed using the GraphPad Prism-8 software. An unpaired or paired, two-tailed student’s *t*-test was performed for the statistical analysis for experiments with two groups. One-way ANOVA with a Bonferroni post-test was used to compare different groups for experiments with more than two groups. Duplicate or triplicate samples were used in the studies. (n) denotes biological replicates, the number of healthy individuals or ALS patients for each experimental condition. The following symbols represent the levels of statistical significance within each analysis: **** (*p*-value <0.0001), *** (*p*-value 0.0001–0.001), ** (*p*-value 0.001–0.01), * (*p*-value 0.01–0.05).

## 3. Results

### 3.1. Genetic Mutational Differences between ALS Patient and Healthy Twin

We analyzed gene mutations of patient and healthy twin using whole genome sequencing (WGS). Among the identified gene mutations, we found a number of them as causative or risk factor for ALS patients (Table 1). ALS patient had five mutations in *TARDBP*, which encodes for the protein *TDP43*. Additionally, *FUS*, encoding an RNA-binding protein, contained a single mutation. Mutations in both genes are reported in a significant number of ALS patients. There were other mutations in ALS patient that were not found in the healthy twin (Table 1). Interestingly, *C9orf72* and *HNRNPA1* mutations, associated with ALS, were not observed in ALS patient but were present in healthy twin, who have not developed any symptoms of ALS. *C9orf72* mutations were shown to account 40% of familial ALS and 7% of sporadic ALS cases [36]. Of interest, the mutation of *PRF1* in ALS patient which encodes perforin might be involved in pathogenesis of the disease. There were 15 additional mutations which were shared between the patient and the healthy twin (Table 1).

### 3.2. Increased NK and B Cell Percentages in the Peripheral Blood of ALS Patients in Comparison to Healthy Individuals

PBMCs of ALS patients and healthy donors were tested to determine the percentages of immune cell subsets. Significantly higher percentages of CD16+ CD56+ NK and CD19+ B cells were seen in PBMCs of ALS patients in comparison to healthy twin or other healthy controls (Appendix A). No differences were seen in the percentages of CD14+ monocytes, CD4+ T, CD8+ T, and NKT cells in ALS patients’ PBMCs in comparison to healthy individuals (Appendix A). These results suggested that ALS patients’ peripheral blood contains higher proportions of NK and B cells in comparison to healthy individuals.

### 3.3. Similar Levels of Cytotoxicity but Significantly Increased IFN-γ Secretion in ALS Patients’ PBMCs when compared to Healthy Individuals

We next used PBMCs against oral squamous carcinoma stem cells (OSCSCs) in a 4 h chromium release assay, and also determined IFN-γ secretion of PBMCs. PBMCs were treated with IL-2, IL-2 + anti-CD16 mAbs, IL-2 + anti-CD3/28 mAbs, and IL-2 + probiotic bacteria sAJ2 before they were used in cytotoxicity assay (Figure 1a–d), or in ELISA (Figure 1e,f), or in ELISpot (Figure 1g,h). Similar levels of cytotoxicity against OSCSCs were induced by PBMCs of ALS patients or healthy individuals (Figure 1a–d). PBMCs of ALS patients secreted higher amounts of IFN-γ when compared to PBMCs of healthy individuals (Figure 1e–h). IL-2 + anti-CD3/28 mAbs, or IL-2 + sAJ2 or IL-2 + anti-PD1-treated ALS patients’ PBMCs secreted significantly increased IFN-γ compared to PBMCs of healthy individuals with the same treatments (Figure 1e–h and Appendix A).

### 3.4. Increased Cell-Mediated Cytotoxicity Was Seen in ALS Patients’ NK Cells in Comparison to Healthy Individuals’ NK Cells

Next, NK cells were treated with IL-2, or IL-2 + anti-CD16 mAbs, or IL-2 + sAJ2 before they were used in a 4 hr chromium release assay (Figure 2a,b), or in ELISA (Figure 2c,d), or in ELISpot (Figure 2e,f). IL-2 + anti-CD16 mAbs-treated ALS patients’ NK cells mediated significantly increased cytotoxicity whereas a slight change in cytotoxicity level was seen in IL-2-treated NK cells (Figure 2a,b). IL-2 + sAJ2-treated NK cells mediated similar levels of cytotoxicity in ALS patients and healthy individuals (Figure 2a,b). Almost similar levels of IFN-γ secretion or IFN-γ spots were seen in ALS patients’ NK cells with IL-2 alone, IL-2 + anti-CD16 mAbs, or IL-2 + sAJ2 treatments (Figure 2c–f). Secretions within the ALS NK samples display as both more polyfunctional and with higher secretion frequency than healthy NK samples at the single cell levels (Figure 2g). On a single cell analysis using Isoplexis platform we observed significant increases in granzyme B and perforin in relation to both signal intensity and heat map for NK cells obtained from ALS patients in comparison to healthy individuals when NK cells were treated with IL-2 + anti-CD16 mAbs (Figure 2h,i). IL-2 + anti-CD16-treated NK samples have 1213 single cells with 1.9% secreting granzyme B, and 4.9% secreting perforin which were significantly higher than those obtained from NK cells obtained from healthy donors (Figure 2h). 1189 single cells were analyzed for IL-2 + sAJ2-treated NK cells from ALS patients which exhibited 1.2% secreting granzyme B, and 4.1% secreting perforin. Although on average the levels are higher in IL-2 + sAJ2-treated NK cells from ALS patients in comparison to healthy donors, they did not achieve statistical significance (Figure 2h).

We next determined the ability and extent of supercharging of NK cells between ALS and healthy individuals using osteoclast-mediated NK expansion methodology [37], and found increased numbers of NK cells at early expansion days which then decreased after day 12 of expansion in ALS patients (Appendix A). Similarly, increased IFN-γ secretion was observed in ALS patients’ NK cells at the early expansion period but it became similar to those obtained from healthy individuals after day 12 of expansion (Appendix A). These findings indicated that increased numbers and activation of NK cells can be seen at earlier stages of expansion in NK cells from ALS patients.

### 3.5. Monocytes Induced Increased Cell-Mediated Cytotoxicity and Secretion of IFN-γ by NK Cells

We next investigated NK cells and monocytes’ interaction. IL-2 + anti-CD16 mAbs-treated NK cells of ALS patients and healthy individuals were cultured with either autologous or allogeneic monocytes at 1:1 ratio overnight followed by measurements of NK cell-mediated cytotoxicity and IFN-γ secretion. Significantly higher NK cell-mediated cytotoxicity was observed when monocytes from ALS patients were cultured with either a healthy twin or other healthy donor or in an autologous manner, although the highest increase in cytotoxicity was seen when patient NK cells were cultured with autologous monocytes (Figure 3a and Appendix A). Similar trends could be seen for the induction of IFN-γ in which patient monocytes has significant activating capacity on the two healthy donors’ NK cells when compared to monocytes from healthy individuals (Figure 3b and Appendix A). Even though healthy monocytes could increase IFN-γ secretion by the patient NK cells when autologous NK cells with patient monocytes were cultured, the levels remained low (Figure 3b and Appendix A). Overall, patient NK cells were able to mediate higher cytotoxicity and secretion of IFN-γ when cultured with autologous monocytes as compared to monocytes obtained from either healthy twin or unrelated healthy individuals.

### 3.6. Increased Effector Memory and Secretion of IFN-γ by ALS Patients’ CD8+ T Cells in Comparison to Healthy Individuals’ CD8+ T Cells

CD4+ and CD8+ T cells of healthy individuals and ALS patients were treated with IL-2, IL-2 + anti-CD3/28 mAbs, or IL-2 + sAJ2 before they were used in ELISA (Figure 4a–c), or in ELISpot (Figure 4d,e). Increased levels of IFN-γ secretion was observed in IL-2-treated CD4+ T cells of ALS patients, whereas IL-2 + anti-CD3/CD28 or IL-2 + sAJ2 treated CD4+ T cells secreted similar levels of IFN-γ secretion in ALS patients and healthy individuals (Figure 4a). In contrast, significantly higher IFN-γ secretion in ALS patients’ CD8+ T cells was observed in all tested conditions including those performed by Luminex analysis (Figure 4b–e, Figure 5, Appendix A). The levels of other pro-inflammatory cytokines, chemokines and growth factors were increased in the cultures of CD8+ T cells from ALS patients (Appendix A). Surface markers of CD8+ T cells were measured using flow cytometry, and increased surface expressions of CD28 and CCR7 on ALS patients’ CD8+ T cells were found (Figure 4f,g). Decreased surface expressions of IFN-γ receptors were also seen on ALS patients’ CD8+ T cells (Figure 4f,g). Significantly higher granzyme B and MIP-1b secretion frequency in CD8+ T cells of ALS patients was seen as compared to those from the healthy controls. 21.4% of ALS1′s single cells and 28.7% of ALS2′s single cells secreted granzyme B in comparison to only 7.6% of healthy control’s single cells secreting granzyme B. In addition, 3.9% of ALS1′s single cells and 6.5% of ALS2′s single cells secreted MIP-1b in comparison to only 1.8% of healthy control’s single cells secreting MIP-1b (Figure 5a). Heatmap graphs exhibited increased monofunctional and polyfunctional CD8+ T cells in ALS patients (Figure 5b). CD8+ T cells from ALS patients demonstrate higher percentages of single cells secreting granzyme B or MIP-1b, and higher percentages of single cells co-secreting granzyme B with MIP-1b (Figure 5c). These results suggested highly elevated function of ALS patients’ CD8+ T cells.

### 3.7. Increased Inflammatory Cytokines in the Serum of ALS Patients in Comparison to Healthy Individuals

Serum was harvested from peripheral blood. Significantly increased secretion of IFN-γ, TNF-α, IL-13, IL-17a, IL-10, IL-23, IL-12p70, and also of MIP-3a, GCSF, IL-10, RANTES, VEGF were observed in ALS patients’ serum in comparison to healthy individuals (Figure 6 and Appendix A). Significantly decreased secretion of Frantalkine, ITAC, and Eotoxin was seen in the serum of ALS patients in comparison to healthy individuals (Figure 6 and Appendix A). Slight modulation was seen in other secreted factors as shown in Figure 6 and Appendix A. These results suggested increased inflammatory cytokines in the serum of ALS patients.

### 3.8. Supernatants Obtained from ALS Patients’ CD8+ T Cells Induced Higher Cell Death and Differentiation of Epithlial Tumor

CD8+ T cells were treated with IL-2 + anti-CD3/28 mAbs overnight before supernatants were harvested and used in the treatment of OSCSCs. lower numbers of tumor cells attached to culture plates were seen when the tumor cells were treated with supernatants from the ALS patients’ CD8+ T cells (Figure 7a and Appendix A). Increased percentages of dead tumor cells were seen in the flow cytometric analysis of patients’ supernatant-treated tumor samples when compared to those treated with healthy individuals’ supernatants (Figure 7b). Increased tumor differentiation was also observed as indicated by higher CD54 surface expression in tumor samples treated with the supernatants of ALS patients’ CD8+ T cells (Appendix A). The increase in tumor differentiation and elevation on surface CD54 expression was shown in our previous studies [25]. These results suggested that CD8+ T cells from ALS patients exhibit a higher potential to induce death and/or differentiation of tumor cells.

### 3.9. Increased Regulatory CD4+ T Cell Subsets in ALS Patients’ PBMCs in Comparison to Healthy Individuals’ PBMCs

We characterized subpopulations of CD4+ T cells in PBMCs, and found increased regulatory T cell (CD4+ CD25+ Foxp3+) percentages in ALS patients’ PBMCs (Figure 8a,d). We then induced differentiation of CD4+ T cells. During the differentiation process, we observed lower cell counts of T-regulatory (Treg) cells in CD4+ T cell cultures of ALS patients (Figure 8e and Appendix A), however, higher secretion levels of IL-10, IFN-γ, and TNF-α were seen by ALS patients’ Treg cells (Figure 8f,g), indicating that either the function or percentages of T reg cells are within the normal or even higher when compared to healthy individuals.

### 3.10. Weekly NAC Injections in ALS Patients Decreased Inflammatory Cytokines in Peripheral Blood except for IFN-γ, TNF-α, IL-17a, and GMCSF

In order to validate the effect of NAC on peripheral blood-derived serum secreted factors, we determined the levels of these factors before and after NAC injection. Significantly decreased levels of IL-10, IL-12p70, IL-13, IL-1b, IL-21, IL-4. IL-23, IL-7, ITAC, Fractalkine, MIP-1a, and MIP-b, and slightly decreased levels of IL-2, IL-5, IL-6, IL-8, and MIP-3a were seen in ALS patients after NAC supplementation (Figure 9a). However, the elevated secretion of IL-17a, TNF-α, IFN-γ, and GMCSF as seen in serum samples before NAC injections were still high after NAC injections (Figure 9 and Appendix A), suggesting the dominant release and function of these cytokines under NAC treatments.

### 3.11. Longitudinal Analysis of CD8+ T Cell Mediated IFN g Secretion from ALS Patient as Compared to Those of the Healthy Identical Twin

We analyzed secretion of IFN-γ by the CD8+ T cells from the patient and his identical twin from March of 2019 to October of 2021 as shown in Figure 10. The levels of IFN-γ secretion remained higher from patient derived CD8+ T cells when compared to those from the healthy twin, with the exception of a few time points in which it coincided with the previous treatments he received. Of 20 assessments, 16 exhibited higher secretion of IFN-γ by the patient derived CD8+ T cells when compared to those from the healthy control. Preceding the time points of 19 September 2019 and 20 October 2019 in which we saw similar levels of IFN-γ secretion between the patient and his healthy twin, the patient received 3 sets of NAC infusions on 1 August 2019, 26 August 2019 and 28 August 2019 in which higher secretion of IFN-g was observed by the patient derived T cells. It appeared that 3 sets of NAC infusions were necessary before we could see a decrease in IFN-γ secretion by the patient derived CD8+ T cells. The next decrease in IFN-g secretion was seen on 21 January 2020 for which the patient had received anti-TNF-α therapy on 5 December 2019 and stem cell injection on 27 December 2019. On 25 September 2020 and 5 October 2020, the patient started taking the pentoxifylline and Amylyax combination, and on 10 November 2020, we observed another decrease in CD8+ T cell-mediated secretion of IFN-γ from the patient. After 16 August 2021, the patient underwent T-reg therapy twice until his death on December 2021 (Figure 10a).

## 4. Discussion

The role of CD8+ T cells in the pathogenesis of ALS has not been clearly established previuosly. As stated in the introduction, CD8+ T cells were found in the spinal cord of ALS mice and ALS patients, and were suggested to contribute to the pathogenesis of ALS by lysis of motor neurons through MHC-class I complex recognition [12]. Although the studies in mice with one dominant mutation are very timely and important, they may not completely represent the human disease in which many gene mutations have been implicated. In addition, we have previously shown that mutations or deletion of many cellular genes are involved in the activation of NK cells [38]. Indeed, several gene mutations were found to be associated with ALS, however, their contributing role in the pathogenesis of ALS is largely unknown. It is also unknown whether mutations in a few dominant genes are the cause of the disease or if multiple genetic abnormalities are responsible for the progression of the disease. Our studies suggest that several mutations in the dominant genes may be responsible for the disease manifestations since, in our preliminary studies, two dominant genes of *TARDBP* and *FUS* were found to be mutated in the ALS patient and not in his healthy identical twin, whereas the healthy identical twin had a mutation of *C9orf72* which has been shown to be associated with the familial form of the disease (Table 1). Although there are differences in the gene mutations between the patient and the healthy identical twin, they also shared a number of gene mutations (Table 1). Therefore, it is possible that other factors are involved in combination with the genetic abnormalities for the disease manifestation and progression. Alternatively, gene mutations in *TARDBP* and *FUS* may be sufficient and necessary for disease manifestations since these mutations were seen in many ALS patients. Of interest, is the observation of *PRF-1* gene mutation in the ALS patient (Table 1), since perforin is highly upregulated in killer T cells and NK cells, and therefore, its mutation may play a role in the pathogenesis of the ALS disease.

In this paper, we studied the function of different immune subsets in ALS patients to determine which main subsets may be contributing to disease induction and progression. Our studies are significant since the function of immune cells from patient were not only compared to other healthy individuals but also it was assessed in genetically identical healthy twin siblings.

Our laboratory has previously reported the list of cellular genes that when deleted or decreased in tumors could augment NK cell function during their interaction with tumor cells [39]. For example, the deletion of *NF-κB* in tumors was found to increase NK cell-mediated cytotoxicity and secretion of IFN-γ [40], and also it resulted in auto-immunity and inflammation in vivo [39]. In addition, conditional knockout of *STAT3* in hematopoietic cells was found to result in the induction of colitis in mice due to chronic gut inflammation [41]. Knock-down of *CD44* in breast and melanoma tumors, and targeted knock-down of *COX2* in non-transformed healthy myeloid cells and mouse embryonic fibroblasts were also able to increase the numbers and functional activation of NK cells [29,39,42,43]. Moreover, mutations in *RAG* gene is known to activate NK cells in patients [44]. It is also likely that gene mutations in ALS sets up the conditions for expansion and functional activation of NK cells and likely CD8+ T cells. Indeed, this study showed that percentages of NK cells were increased in ALS patients, and unlike those obtained from either the identical twin or those of the other healthy controls, their cytotoxic function did not decrease when treated with IL-2 + anti-CD16 mAbs (Figure 2a,b). Indeed, we have previously shown that treatment with IL-2 and anti-CD16mAb significantly decreases the cytotoxic function of NK cells in healthy individuals while increasing the secretion of IFN-γ, a concept coined as split anergy in NK cells [23]. In addition, we have also observed increased Granzyme B and perforin in NK cells treated with IL-2 + anti-CD16 mAbs from ALS patients in comparison to his identical twin on a single cell level (Figure 2h,i). Although on average higher induction of IFN-γ spots could be seen with IL-2 + anti-CD16 mAbs treatment of NK cells from ALS patients, the results did not reach statistical significance. No significant differences in IFN-γ secretion was observed between ALS patients and those of healthy individuals (Figure 2e,f). Moreover, sAJ2-activated NK cells increase IFN-γ spots or secretion in ALS patients more than in either healthy identical twin or in other healthy controls. Thus, there is an elevation of patient-derived NK function under certain treatment modalities.

When the NK function was determined after their culture with autologous monocytes, significantly higher NK cell-mediated cytotoxicity was observed when monocytes from ALS patients were cultured with either genetically identical twin or other healthy donors or in an autologous manner, although the highest increase in cytotoxicity was seen when patient NK cells were cultured with autologous monocytes. Similar trends could be seen for the induction of IFN-γ in which patient monocytes had significant activating capacity on the two healthy donors’ NK cells when compared to monocytes from healthy individuals. Even though monocytes from healthy individuals could increase patients’ NK cell-mediated IFN-γ secretion, when autologous NK cells with patient monocytes were cultured, the levels remained low suggesting potential regulation of IFN-γ secretion on an autologous basis due to the factors that are not completely understood yet. However, patient monocytes were able to increase IFN-γ secretion by allogeneic NK cells from healthy individuals and the levels of IFN-γ release were higher when compared to those from autologous cultures of patient NK cells with monocytes. Overall, patient NK cells were able to mediate higher cytotoxicity when cultured with autologous monocytes as compared to those obtained from either healthy twin or unrelated healthy control.

Primary NK cells treated with IL-2 + anti-CD16 mAbs and cultured with osteoclasts increases expansion, cytotoxicity and induction of IFN-γ secretion significantly in patients similar to those seen with healthy individuals, providing the rationale for coining them as super-charged NK (sNK) cells. We have previously shown that sNK cells preferentially target and kill activated CD4+ T cells but they expand and activate CD8+ T cells, and also give rise to memory effector CD8+ T cells [45]. Therefore, activated NK cells in ALS patients may have a significant effect on the expansion of CD8+ T cells. The levels of IFN-γ spots and secretion were higher in NK cells in comparison to CD8+ T cells per cell basis [45]. Indeed, mouse studies have shown higher expression of TLR2, TLR3, TLR4, and TLR7 receptors on NK cells in comparison to CD8+ T cells [46].

The highest levels of IFN-γ spots or secretion were found in CD8+ T cells treated with anti-CD3/CD28 antibodies, and the levels were significantly higher in ALS patients’ CD8+ T cells in comparison to healthy controls. However, in all different types of treatments of CD8+ T cells, increased IFN-γ spots and secretion were observed in ALS patients when compared to healthy controls. CD8+ T cells of ALS patients had a higher surface expression of CD28 and CCR7, but a lower surface expression of IFN-γ receptor α and β chains. It is possible that continuously increased secretion of IFN-γ in patients allows binding of IFN-γ to its receptors on CD8+ T cells and therefore, competes with the binding of IFN-γ receptor antibody. Alternatively, activated CD8+ T cells shed their IFN-γ receptors and therefore, are not able to bind to the secreted IFN-γ and are not regulated by the secretion of IFN-γ to inhibit further activation. These scenarios are under investigation in our laboratory.

The addition of supernatants obtained from anti-CD3/CD28 activated CD8+ T cells from ALS patients increased differentiation antigens such as CD54 on OSCSCs and mediated higher cell death of these cells when compared to those obtained from either healthy identical twin or other healthy individuals. These experiments suggest similar mechanisms of action on motor neurons by highly activated CD8+ T cells in ALS patients.

We have previously shown that both IFN-γ and TNF-α are important in the induction of differentiation of tumor cells as well as healthy stem cells [25]. Increased differentiation of the motor neuron stem cells by IFN-γ and TNF-α secreted by the activated CD8+ T cells can potentially increase their targetability by the CD8+ T cells since increased differentiation is able to increase MHC-class I and peptide-mediated lysis by the CD8+ T cells, as suggested by the murine data [12].

In addition to increased IFN-γ and TNF-α, CD8+ T cells from patients were found to have much higher levels of granzyme B on single cell level suggesting a potential mechanism of CD8+ T cell-mediated killing of motor neurons similar to those shown in the mouse model of ALS [12].

We followed the patient and his identical twin longitudinally to determine whether any of the treatments received would change the function of CD8+ T cells. The time points preceding 19 September 2019 and 20 October 2019, the patient started receiving NAC infusions on 1 August 2019, 26 August 2019 and 28 August 2019 and on 19 September 2019 and 20 October 2019 we observed decreases in CD8+ T cell mediated IFN-γ secretion from the patient. Even though the patient continued receiving regular NAC infusions, the decrease did not last and the patient continued exhibiting increased IFN-γ secretion when compared to his healthy twin brother (Figure 10), suggesting that tolerance might have been induced. The next decrease was seen on 21 January 2020 and the patient had received anti-TNF-α therapy on 5 December 2019 and stem cell injection on 27 December 2019 and we observed similar levels of IFN-γ from CD8+ T cells between the patient and healthy twin on 21 January 2020. Although we observed a decrease in IFN-γ secretion by the patients’ CD8+ T cells, the effect did not last and the levels of IFN-γ continued rising, suggesting the short-lived nature of these treatments. On 10 November 2020 we saw a decrease in CD8+ T cell mediated secretion of IFN-γ from the patient. On 25 September 2020 and 5 November 2020 patient started taking pentoxifylline and Amylyax combination. Amylyex is a combination of two orally administered small molecules, tauroursodeoxycholic acid (TUDCA) and sodium phenylbutyrate, which the suggested mode of action is to protect nerve cells from damage to the endoplasmic reticulum and mitochondrial-dependent neuronal degeneration pathways in ALS. Pentoxifylline is shown to inhibit the function of TNF-α [47]. Although the combination of these two compounds might have decreased the IFN-γ secretion from the patients’ CD8+ T cells, the inhibition was short lived once again since the levels continued rising after 5 October 2020. After 16 August 2021 the patient underwent T reg therapy twice until his death on December 2021. The therapy resulted in a continuous decline from the highs of 24 March 2021, but due to his untimely death these experiments were terminated on December 2021. Most of the treatments administered were short lived in the patient likely due to a buildup of tolerance to the treatments. It is currently unclear how tolerance is established in the CD8+ T cell function, however, these studies have provided important directions which will be investigated in the future in an ALS relevant animal model as well as in other patients.

Significantly higher levels of IFN-γ and TNF-α were observed in the serum of ALS patients, correlating with the increased secretion of these cytokines from PBMCs, NK cells, and CD8+ T cells, and indicating the dominant role of these cytokines in the pathogenesis of ALS. Increased serum IFN-γ and TNF-α were seen in the comparison between the ALS patient with genetically identical twin and also between healthy donors and the other ALS patients tested in the study. Significantly elevated levels of IL-17a and IL-10 could also be seen in both group comparisons. Although higher levels of IL12p70 and IL-6 were seen in both groups, they reached significance between the other ALS patients when compared to all other healthy individuals. Similarly, the levels of G-CSF, Eotaxin, IP10, Rantes and VEGF reached significant levels among the other ALS patients when compared to all other healthy individuals. The levels of Fractalkine and MIP3a were significantly higher in serum between the healthy twin and the ALS patient and the reverse was seen with ITAC. These results point to the elevated levels of both pro and anti-inflammatory cytokines in ALS patients and suggest an overall heightened activation of the immune system, in particular highlighting the significance of IFN-γ and TNF-α in the pathogenesis of ALS. Indeed, when the levels of Tregs were determined in patients, higher percentages of Foxp3 populations could be seen in the PBMCs, and when naïve CD4+ T cells were differentiated to Tregs even though decreased numbers of expanded Tregs could be seen in ALS patients, they secreted higher levels of IL-10 when compared to the healthy controls. The level of IL-10 was higher than that of IFN-γ secretion from Tregs. Indeed, in our preliminary experiments, the ratio of IFN-γ to IL-10 is much lower for the patients that have recently been diagnosed and higher for those that have been diagnosed several years ago. It is possible that by lowering the ratio of IFN-γ to IL-10 through the infusion of Treg cells which block NK and CD8+ T cell function we may be able to stop or decrease the progression of the disease in patients that have had the disease for longer periods. Indeed, such patient assessments are undergoing at present in our laboratory.

Among the treatments that some of the ALS patients underwent were infusions of weekly NAC starting in August 2019, and continuing until December 2021. NAC infusions were administered as part of their care by their physicians. When we assessed the serum concentrations of cytokines and chemokines before and after NAC infusions in the patient most cytokines and chemokines and growth factors demonstrated decreased levels with the exception of IFN-γ, TNF-α, and IL-17a which remained higher even after NAC infusions. These results indicated that the two cytokines that are crucial for mediating differentiation of the cells are not controlled by NAC treatment which is shown to block cell death. The ultimate goal of any effective therapy is lowering of IFN-γ and TNF-α in ALS to protect the cells from over-differentiation/activation which could cause increased cell death. In addition, NAC may also cause increases in the survival of auto-reactive CD8+ T cells and exhibit a worst outcome in the patient. However, we did not observe a worsening of the symptoms upon treatment with NAC. Indeed, many of the cytokines and chemokines secreted by the CD8+ T cells were decreased with the exception of IFN-γ, TNF-α, and IL-17a. A decrease in chemokine secretion may contribute to decreased recruitment of T cells to the diseased sites and alleviate the pathologies caused by the CD8+ T cells. Moreover, increasing amounts of IFN-γ and TNF-α secreted by the CD8+ T cells may also exacerbate differentiation of the hyperactive T cells resulting in increased activation induced cell death of CD8+ T cells, all beneficial to the patient. However, increased IFN-γ and TNF-α can also affect motor neurons on an adverse way by over differentiating and causing higher levels of cell death. Thus, the effect of NAC can be multifactorial and further assessments are needed to establish the full spectrum of its function in ALS treatment.

Lack or decreased IFN-γ receptor expression on CD8+ T cells is very important since this cytokine not only is important for the functional activation of the immune cells, but they are also important in limiting the survival and function of NK and CD8+ T cells [48,49,50]. Indeed, IFN-γ is shown to limit its own production by the immune cells [51,52]. Due to the lack of IFN-γ receptors, it is possible that CD8+ T cells survive longer and have more capability to kill motor neurons. We are in the process of delineating the significance of IFN-γ Receptors both on CD8+ T cells and on motor neurons in ALS.

The use of drugs such as cyclophosphamide and azathioprine which inhibit DNA replication and cell proliferation, and prednisone which serves as an anti-inflammatory and immunosuppressant has not yielded successful control of the disease indicating that general blocking of the immune system may not be appropriate treatment strategies, and that we should look for a more targeted therapy which could block the aggressive functions of NK and CD8+ T cells in terms of both direct killing as well as increased secretion of IFN-γ and TNF-α which could potentially over differentiate and induce increased cell death as seen in this report.

For the sake of brevity and keeping the size of the manuscript manageable, and at the same time to present the key experiments, we moved a great number of important results to the Appendix A. In addition, the discussion had to be shortened and therefore, many important and key findings had to be discussed briefly, which we hope to expand in our future publications and reviews. In addition, to present the patient data without confounding factors accounting for variabilities seen from day to day experiments, and/or the methodologies used we opted to demonstrate single representative experiments as well as compiled patient data. We have also performed analysis of the patients longitudinally and not just for one time point to observe whether the increase in CD8+ T cells is a consistent pattern and not sporadic occurrences. Therefore, the compiled data included repeated assessments of some patients at different time intervals as well as patients with single assessments. Overall, our studies are highly important and point to the existence of aggressive CD8+ T cells; functions of which remain high throughout the disease progression, unless an effective therapeutic strategy can be designed to bring their function under control continuously and not for a short duration of time as seen in our studies. Without the control of these cells any attempts in the regenerative treatment of ALS patients may be ineffective.

## Figures and Tables

**Figure 1 cells-11-03431-f001:**
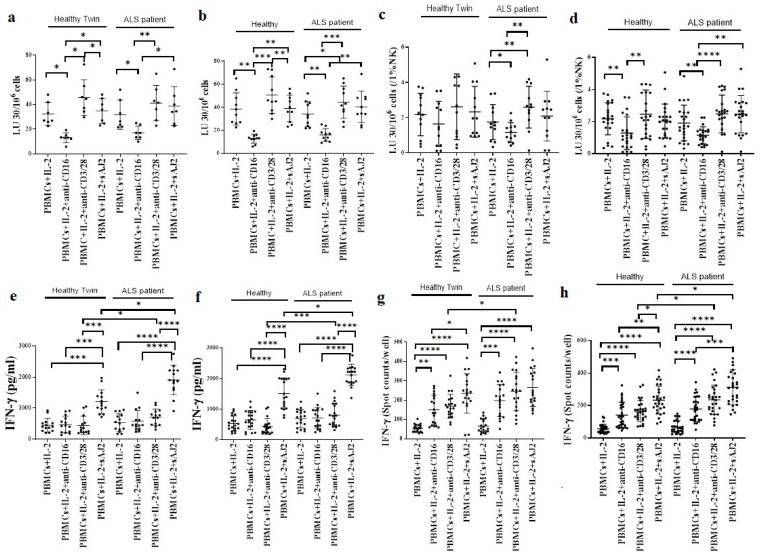
Significant increased IFN-γ secretion but not NK cell-mediated cytotoxicity was seen in PBMCs of ALS patients. PBMCs of healthy individuals and ALS patients were isolated from peripheral blood as described in Section 2. PBMCs were treated with IL-2 (1000 U/mL) or with a combination of IL-2 (1000 U/mL) and anti-CD16 mAbs (3 µg/mL) or IL-2 (1000 U/mL) and anti-CD3/28 antibody (25 µL/mL) or IL-2 (1000 U/mL) and sAJ2 (PBMC:sAJ2, 1:20) for 18 h before they were used in standard 4 h ^51^Cr release cytotoxicity assay against OSCSCs. The lytic units (LU) 30/10^6^ cells were determined using the inverse number of PBMCs required to lyse 30% of OSCSCs ×100 ((**a**) (*n* = 7), (**b**) (*n* = 10)). LUs of Figure 1a ((**c**) (*n* = 12)) and Figure 1b ((**d**) (*n* = 23)) were used to determine LUs per 1% NK cells using percentages of CD16+ cells in PBMCs obtained by flow cytometric analysis. PBMCs were treated as described in Figure 1a, 18–20 h after treatments, the supernatants were harvested to determine IFN-γ secretion using single ELISA ((**e**) (*n* = 14), (**f**) (*n* = 18)). PBMCs were treated as described in Figure 1a, 18–20 h after treatments, the number of cells secreting IFN-γ were determined as spot counts using ELISpot assay ((**g**) (*n* = 19, (**h** (*n* = 28)). **** (*p*-value < 0.0001), *** (*p*-value 0.0001–0.001), ** (*p*-value 0.001–0.01), * (*p*-value 0.01–0.05).

**Figure 2 cells-11-03431-f002:**
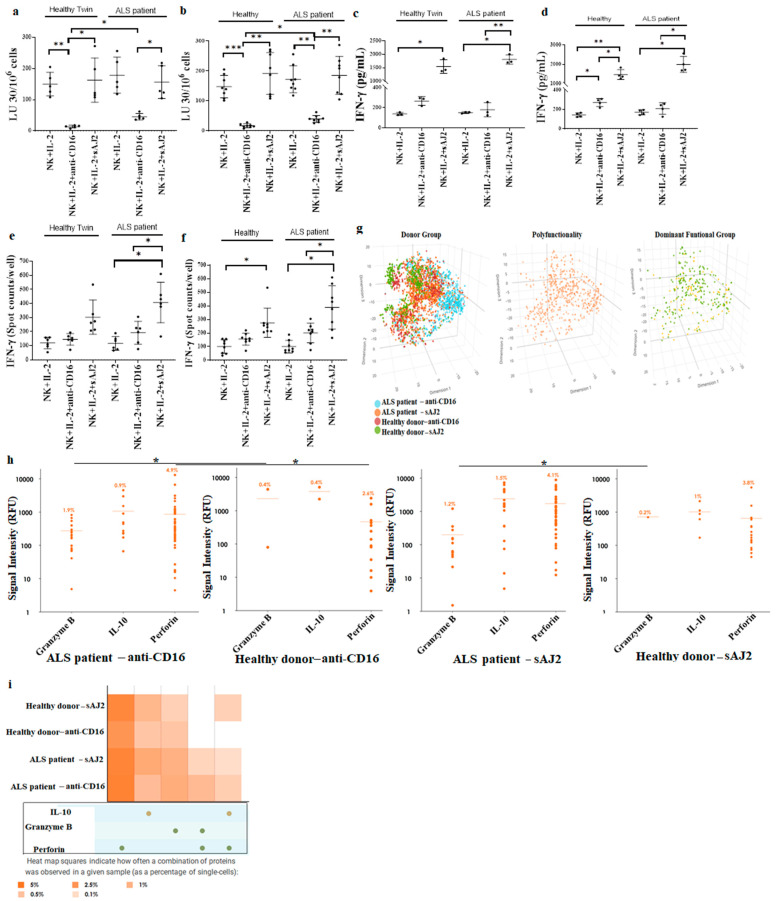
Increased NK cell-mediated cytotoxicity in IL-2 and anti-CD16mAb-treated NK cells from ALS patients. NK cells of healthy individuals and ALS patients were isolated from PBMCs as described in Section 2. NK cells (1 × 10^6^ cells/mL) were treated with IL-2 (1000 U/mL) or with a combination of IL-2 (1000 U/mL) and anti-CD16 mAbs (3 µg/mL) or IL-2 (1000 U/mL) and sAJ2 (NK:sAJ2, 1:2) for 18 h before they were used in a standard 4 h ^51^Cr release cytotoxicity assay against OSCSCs. The lytic units (LU) 30/10^6^ cells were determined using the inverse numbers of NK cells required to lyse 30% of OSCSCs ×100 ((**a**) (*n* = 5), (**b**) (*n* = 8)). NK cells were treated as described in Figure 2a, 18–20 h after treatments, the supernatants were harvested to determine IFN-γ secretion using single ELISA ((**c**) (*n* = 3), (**d**) (*n* = 4)). NK cells were treated as described in Figure 2a, 18–20 h after treatments, the number of cells secreting IFN-γ were determined as spot counts using ELISpot assay ((**e**) (*n* = 6), (**f**) (*n* = 8)). 3D t-SNE visualizations demonstrate ALS patients’ NK cells do not co-cluster with healthy donors’ NK cells. Each dot in the t-SNE scatterplot corresponds to a single cell, color coded by donor group, polyfunctionality or a characteristic of the cell’s secretion profile. t-SNE visualization of all single-cell chambers can distinguish subsets of donor groups. Removing non-secreting cells from t-SNE visualization allows for selection of polyfunctional cells. Further categorization of secreting cells into dominant functional groups (**g**). NK cells from ALS patients and healthy individuals were treated with IL-2 in combinations with anti-CD16 mAb or sAJ2, and secretion frequency of granzyme B, IL-10 and perforin were measured at single cell level. NK cells from ALS patients treated with IL-2+ anti-CD16 has 1213 single cells with 1.9% secreting Granzyme B, 0.9% secreting IL-10 and 4.9% secreting perforin. NK cells from healthy donors treated with IL-2+anti-CD16 mAb has 509 single cells with only 0.4% secreting Granzyme B, 0.4% secreting IL-10 and 2.4% secreting perforin. Granzyme B and perforin’s signal intensities from all single cells are significantly different between ALS patients and healthy donors after IL-2+anti-CD16 mAb treatment. NK cells from ALS patient treated with IL-2+sAJ2 has 1189 single cells with 1.2% secreting Granzyme B, 1.5% secreting IL-10 and 4.1% secreting perforin. NK cells from healthy twin treated with IL-2+sAJ2 has 480 single cells with only 0.2% secreting Granzyme B, 1% secreting IL-10 and 3.8% secreting perforin. However, Granzyme B, IL-10 and perforin’s signal intensities from all single cells are not significantly different between ALS patients and healthy donors after IL-2+sAJ2 treatment. (**h**). Heatmap graphs show increased polyfunctional NK cell subsets within ALS patients. Heatmaps compare the percentage of single NK cells secreting various monofunctional and polyfunctional groups across multiple samples. Secretions within the ALS samples display as both more polyfunctional and with higher secretion frequency than healthy samples. The combined secretion of Granzyme B and perforin is unique to ALS samples (**i**). *** (*p*-value 0.0001–0.001), ** (*p*-value 0.001–0.01), * (*p*-value 0.01–0.05).

**Figure 3 cells-11-03431-f003:**
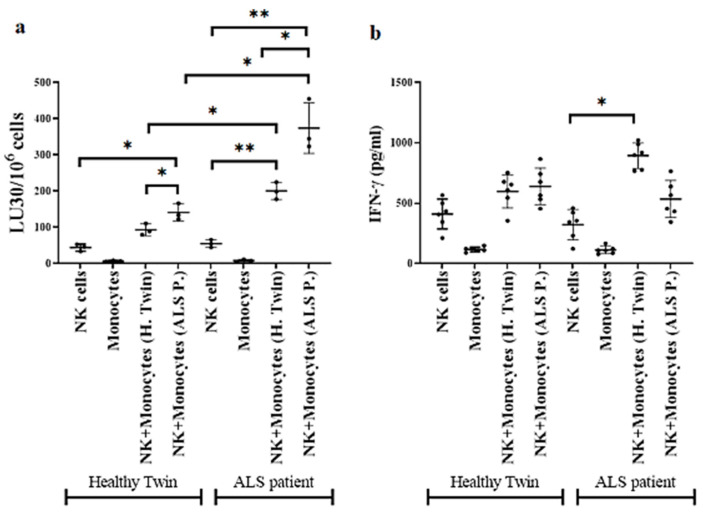
Autologous and allogeneic monocytes from ALS patients and those of healthy individuals increase NK cell-mediated cytotoxicity and secretion of IFN-g. NK cells and monocytes from ALS patients and healthy individuals were isolated from PBMCs as described in Section 2. NK cells were treated with a combination of IL-2 (1000 U/mL) and anti-CD16 mAbs (3 µg/mL). A crisscross NK cells and monocyte co-cultures were performed with 3 sets of NK cells with allogeneic and autologous monocytes. NK cell- mediated cytotoxicity were measured 18 h after co-culture using standard 4 h ^51^Cr release assay against OSCSCs. The lytic units (LU) 30/10^6^ cells were determined using inverse number of NK cells needed to lyse 30% of target cells OSCSCs ×100 (*n* = 3) (**a**). NK and monocyte co-cultures were performed as described in (**a**). After 18 h of co-culture, supernatants were harvested and used in ELISA to measure IFN-γ secretion (*n* = 6) (**b**). ** (*p*-value 0.001–0.01), * (*p*-value 0.01–0.05).

**Figure 4 cells-11-03431-f004:**
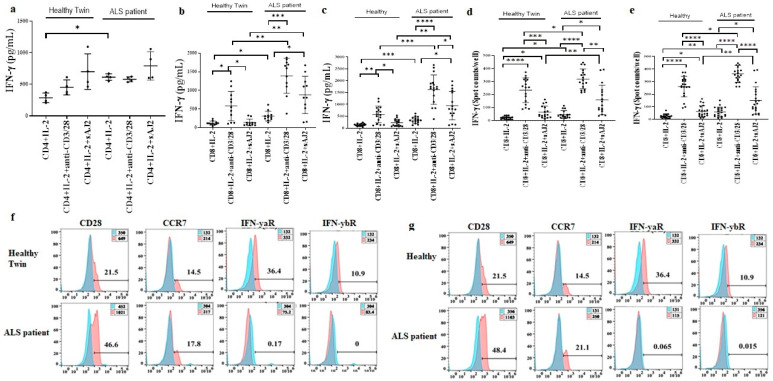
Increased effector memory and secretion of IFN-γ by CD8+ T cells from ALS patients in comparison to healthy individuals. CD4+ and CD8+ T cells of healthy individuals and those from ALS patients were isolated from PBMCs as described in Section 2. CD4+T cells (1 × 10^6^ cells/mL) were treated with IL-2 (100 U/mL) or with a combination of IL-2 (100 U/mL) and anti-CD3/28 antibody (25 µL/mL) or IL-2 (100 U/mL) and sAJ2 (CD4+ T:sAJ2, 1:2) for 18 h before the supernatants were harvested to determine IFN-γ secretion using single ELISA ((**a**) (*n* = 4)). CD8+T cells (1 × 10^6^ cells/mL) were treated with IL-2 (100 U/mL) or with a combination of IL-2 (100 U/mL) and anti-CD3/28 antibody (25 µL/mL) or IL-2 (100 U/mL) and sAJ2 (CD8+ T:sAJ2, 1:2) for 18 h before the supernatants were harvested to determine IFN-γ secretion using single ELISA ((**b**) (*n* = 11), (**c**) (*n* = 17)). CD8+ T cells were treated as described in (**b**), 18–20 h of treatments, the number of cells secreting IFN-γ were determined as spot counts using ELISpot assay ((**d**) (*n* = 18), (**e**) (*n* = 20)). CD8+ T cells of healthy individuals and ALS patients were isolated from PBMCs as described in Materials and Methods. Surface expression of CD28, CCR7, IFN-γαR, and IFN-γβR on CD8+ T cells were analyzed using flow cytometry, IgG2 isotype control antibody was used as control (**f**,**g**). One of five representative experiments is shown in (**f**,**g**). **** (*p*-value < 0.0001), *** (*p*-value 0.0001–0.001), ** (*p*-value 0.001–0.01), * (*p*-value 0.01–0.05).

**Figure 5 cells-11-03431-f005:**
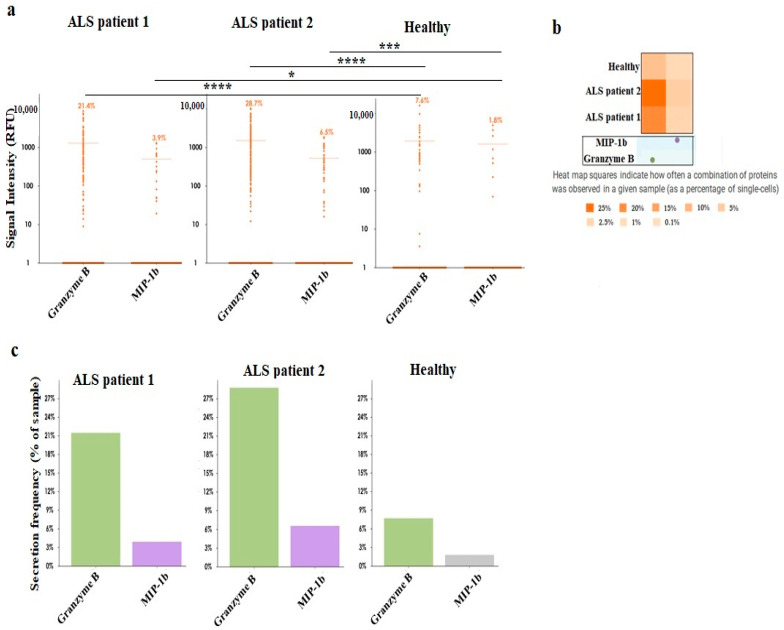
Determining the levels of Granzyme B and MIP-1b in CD8+ T cells of ALS patients and healthy individuals. CD8+ T cells from ALS patients and healthy individuals were treated with anti-CD3/28 antibody and secretion frequency of Granzyme B, and MIP-1b were measured at single cell level. ALS1 sample has 462 single cells; ALS2 sample has 536 single cells; and healthy control sample has 510 single cells. 21.4% of ALS1′s single cells and 28.7% of ALS2′s single cells secreted Granzyme B in comparison to only 7.6% of healthy control’s single cells secreting Granzyme B. In addition, 3.9% of ALS1′s single cells and 6.5% of ALS2′s single cells secreted MIP-1b in comparison to only 1.8% of healthy control’s single cells secreting MIP-1b. Granzyme B’s signal intensities from all single cells among 3 donors are significantly different. MIP-1b’s signal intensities from all single cells between ALS patients are not significantly different but MIP-1b’s signal intensities are significantly different between ALS patients and healthy control (**a**). Heatmap graphs indicates how often a combination of proteins was observed in a given sample. Heatmaps compare the percentage of single cells secreting various monofunctional and polyfunctional groups across multiple samples. ALS samples possess higher percentage of single cells secreting Granzyme B or MIP-1b; and higher percentage of single cells co-secreting Granzyme B with MIP-1b (**b**). CD8+ T cells from ALS patients and healthy individuals were treated with anti-CD3/28 antibody and secretion frequency of Granzyme B, and MIP-1b were measured at single cell level (**c**). One of five representative experiments is shown in (**c**). **** (*p*-value <0.0001), *** (*p*-value 0.0001–0.001), * (*p*-value 0.01–0.05).

**Figure 6 cells-11-03431-f006:**
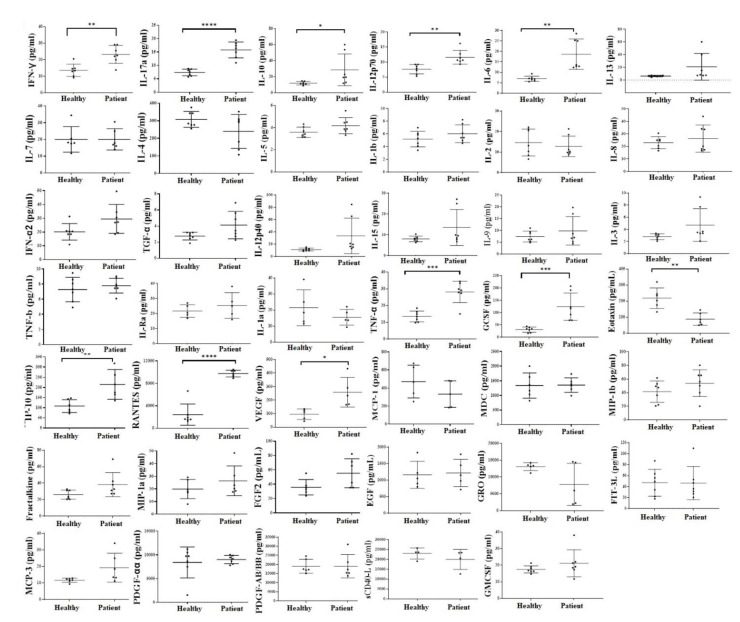
Inflammatory cytokines in the peripheral blood-derived serum of ALS patients in comparison to those from healthy individuals. Sera were obtained from the peripheral blood of healthy individuals and ALS patients, and analyzed for the levels of cytokines, chemokines, and growth factors using a multiplex array kit (*n* = 7). **** (*p*-value < 0.0001), *** (*p*-value 0.0001–0.001), ** (*p*-value 0.001–0.01), * (*p*-value 0.01–0.05).

**Figure 7 cells-11-03431-f007:**
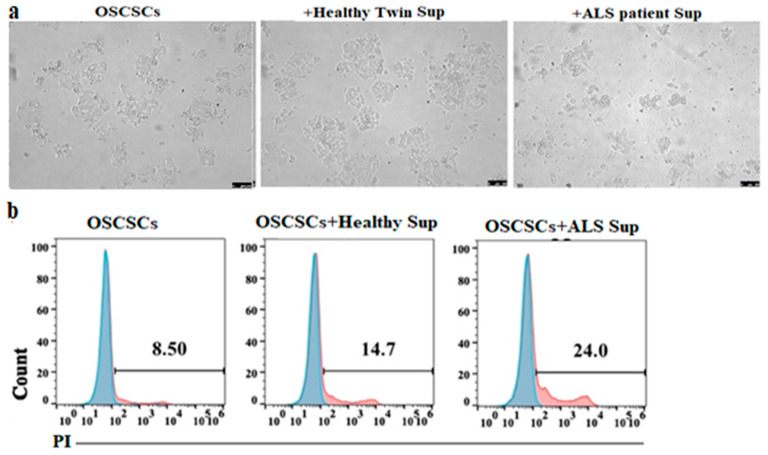
Supernatants from CD8+ T cells from ALS patients induced higher cell death of OSCSCs. CD8+ T cells of healthy individuals and ALS patients were isolated from PBMCs as described in Section 2. CD8+T cells (1 × 10^6^ cells/mL) were treated with a combination of IL-2 (100 U/mL) and anti-CD3/28 antibody (25 µL/mL) before the supernatants were harvested to determine IFN-γ secretion using single ELISA. Supernatants containing IFN-γ from the healthy individuals and ALS patients’ CD8+ T cells were added to the same numbers of OSCSCs cultured in the plates for 2 days. On day 2, pictures of culture plates were taken using inverse microscope (**a**), OSCSCs were detached and the levels of cell death was determined using propidium iodide staining by flow cytometry (**b**). One of six representative experiments is shown in the figure.

**Figure 8 cells-11-03431-f008:**
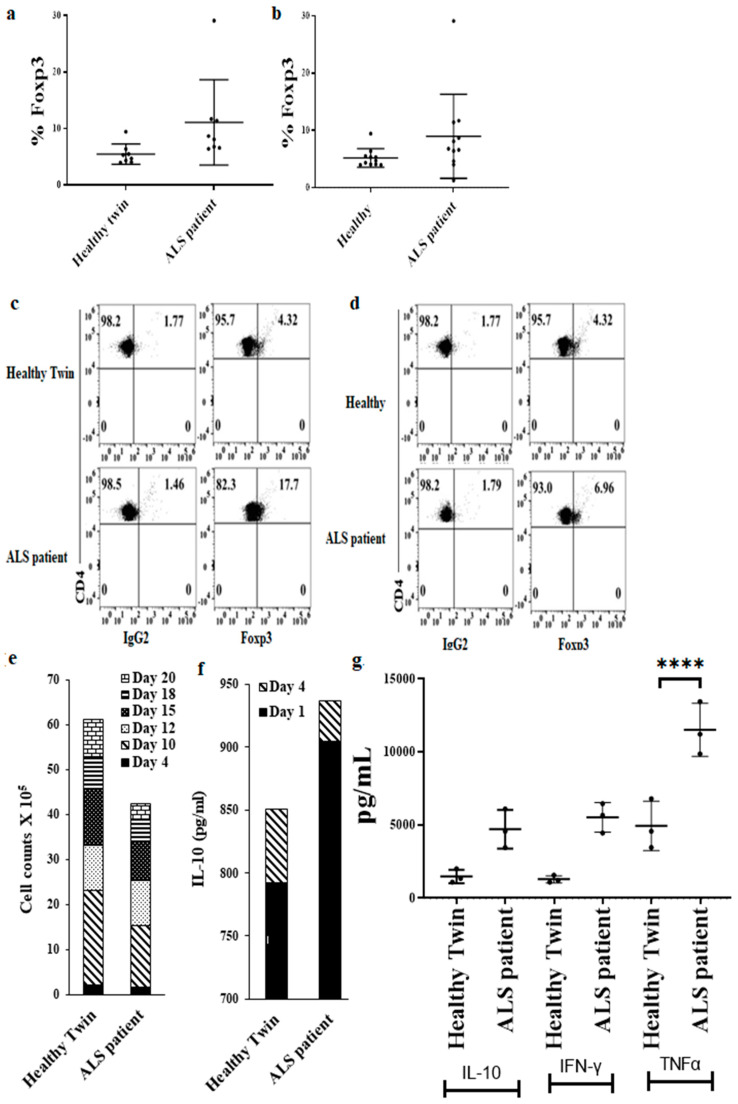
Increased regulatory CD4+ T cells in PBMCs of ALS patients in comparison to healthy individuals. PBMCs of healthy individuals and ALS patients were isolated from peripheral blood as described in Section 2. PBMCs (2 × 10^5^ cells) were used to determine the percentages of Foxp3 using flow cytometric analysis. IgG2 isotype control antibody was used as control ((**a**) (*n* = 8), (**b**) (*n* = 11)). PBMCs (2 × 10^5^ cells) were used to determine the percentages of CD4+Foxp3+ using flow cytometric analysis. IgG2 isotype control antibody was used as control (**c**,**d**). CD4+ T cells of healthy individuals and ALS patients were isolated from PBMCs as described in Section 2. For Treg differentiation, naïve CD4+ T cells (1 × 10^6^ cell/mL) were cultured with ImmunoCult-XF T cell expansion medium supplemented with Immunocult human CD3/CD28 T cell activator (25 µL/mL). On days 4, 10, 12, 15, 18, and 20, the cell counts were performed manually using microscopy (**e**). CD4+ T cells were treated as described in (**e**), on days 1 and 4, the supernatants were harvested to determine IL-10 secretion using single ELISA (**f**). CD4+ T cells were treated as described in (**e**), on day 1, the supernatants were harvested to determine IL-10, IFN-γ, and TNF-α secretion using multiplex assay (*n* = 3) (**g**). **** (*p*-value < 0.0001).

**Figure 9 cells-11-03431-f009:**
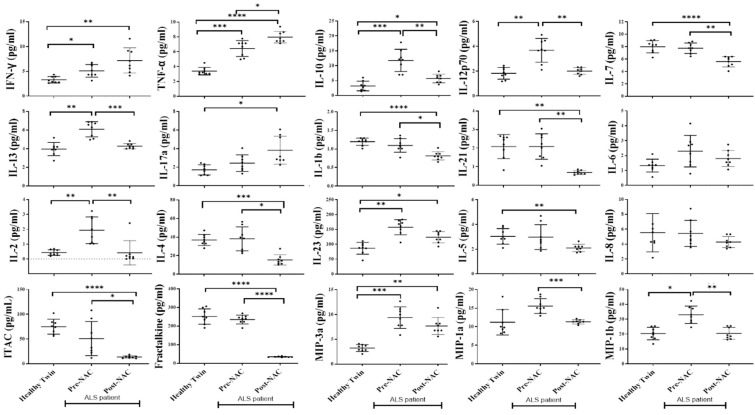
Weekly NAC injections in ALS patients decreased inflammatory cytokines in peripheral blood except IFN-γ, TNF-α, IL-17a and GMCSF. Sera were obtained from the peripheral blood of healthy individuals and ALS patients (before and after NAC injection in patient), and analyzed for the levels of cytokines, chemokines, and growth factors using a multiplex array kit (*n* = 8). **** (*p*-value < 0.0001), *** (*p*-value 0.0001–0.001), ** (*p*-value 0.001–0.01), * (*p*-value 0.01–0.05).

**Figure 10 cells-11-03431-f010:**
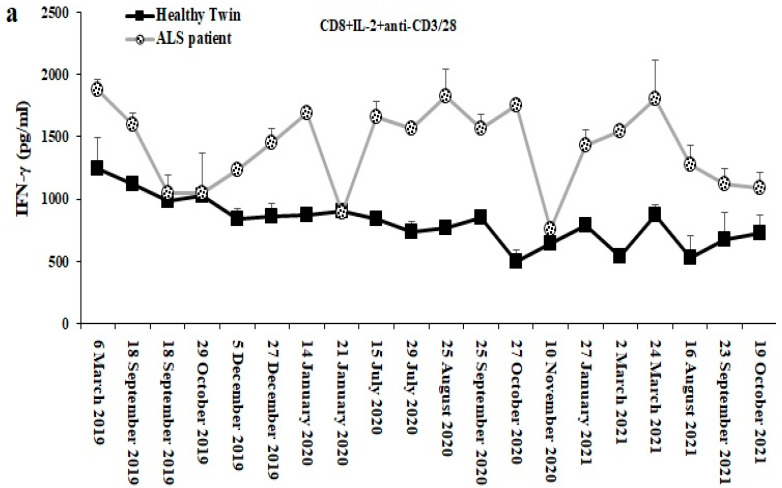
Levels of IFN-γ secretion by CD8+ T cells from ALS patients in comparison to healthy individuals in chronological order. CD8+ T cells of healthy individuals and those from ALS patients were isolated from PBMCs as described in Section 2. CD8+T cells (1 × 10^6^ cells/mL) were treated with a combination of IL-2 (100 U/mL) and anti-CD3/28 antibody (25 µL/mL) for 18 h before the supernatants were harvested to determine IFN-γ secretion using single ELISA (a). The data was obtained at different time points as shown in the figures. Duplicate samples from healthy individual and ALS patient were used at each time point.

**Table 1 cells-11-03431-t001:** List of gene mutations in ALS patient and healthy twin.

Whole Genome Sequencing Analyses		
ALS Patient	Healthy	Shared Gene Mutations
TARDBP	C9orf72	ALS2
ERBB4	HNRNPA1	NEK1
PRF1		PRPH2
ANG		FIG4
SPG11		ELP3
ATXN21		SIGMAR1
FUS		SETX
		OPTN
		ATXN2
		TRPM7
		PFN1
		SARM1
		TAF15
		UNC13A
		NEFH

## Data Availability

This study did not report any data.

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
