# Peer review of "The Potential Role of Cytotoxic Immune Effectors in Induction, Progression and Pathogenesis of Amyotrophic Lateral Sclerosis (ALS)"

_cells, 2022, doi:10.3390/cells11213431_

Round 1

Reviewer 1 Report

In the submitted manuscript Kaur et al. studied different aspects of immune pathogenesis in ALS. They we comprehensively analyzed the phenotype in PBMCs of ALS patients in comparison to either their genetically identical twin or with other healthy donors in order to determine whether correlations could be found with disease progression.

While this is interesting preliminary data, the biggest concerns which limit the impact is that the data about twin are poorly cited and discussed while may be very interesting.

Moreover, is not specify the number of samples used neither in methods nor in results paragraph.

Also, in the results about NAC infusions there are major limitations which the authors should address in the discussion.

Author Response

August 19, 2022

We appreciate the hard work of our reviewers and have addressed their concerns. Below please find point by point response to the reviewers’ comments. We hope that the explanations and significantly modified version of the manuscript is now satisfactory for publication.

Thanks

Dr. Jewett

In the submitted manuscript Kaur et al. studied different aspects of immune pathogenesis in ALS. They we comprehensively analyzed the phenotype in PBMCs of ALS patients in comparison to either their genetically identical twin or with other healthy donors in order to determine whether correlations could be found with disease progression.

While this is interesting preliminary data, the biggest concerns which limit the impact is that the data about twin are poorly cited and discussed while may be very interesting.

Response: We thank the reviewer for the comments and have now added a significant portion to the manuscript regarding the genetic analysis of the ALS patient in comparison to his identical twin brother. We have also expanded both the results and the discussion sections regarding the differences seen between the ALS patient and his healthy twin brother.  We hope the revision is satisfactory.

Moreover, is not specify the number of samples used neither in methods nor in results paragraph.

Response: We apologize for not clearly enumerating the total numbers of experiments we performed.  For the comparison between ALS patient and identical healthy brother we performed 25 independent assessments from June 2018 to Nov of 2020. For other ALS patients compared to age and sex matched healthy donors we had 12 different healthy donors and 7 ALS patients and we performed 37 different assessments for each group from June 2018 to Nov of 2020.

Also, in the results about NAC infusions there are major limitations which the authors should address in the discussion.

Response: We have now included the limitations of NAC infusions in the discussion section as suggested by the reviewer.

Reviewer 2 Report

In this manuscript, the authors describe the immune profile of peripheral blood mononucleated cells from two ALS patients versus controls using a variety of methodologies. While in principle the study is interesting, it is limited to only 2 ALS patients and 2 controls, a number way too small to being able to draw any valid conclusions. There are many additional issues with this manuscript. There is no clinical and genetic description of the patients and controls, the description of the methods is incomplete (particularly regarding the NAC treatment), it is unclear it the IRB approval also covered NAC injections, many of the data presented are not statistically significant, it is unclear what the n represents in most experiments (number of technical replicates or biological replicates?), and some of the data seem to be based on only one patient and have no standard deviations (only one experiment? e.g. Fig 7 and 8). As is, this manuscript is not suitable for publication in my view. 

Author Response

We appreciate the hard work of our reviewers and have addressed their concerns. Below please find point by point response to the reviewers’ comments. We hope that the explanations and significantly modified version of the manuscript is now satisfactory for publication. 

Thanks

Dr. Jewett

In this manuscript, the authors describe the immune profile of peripheral blood mononucleated cells from two ALS patients versus controls using a variety of methodologies. While in principle the study is interesting, it is limited to only 2 ALS patients and 2 controls, a number way too small to being able to draw any valid conclusions.

Response: We apologize if we have not clearly indicated the number of healthy individuals and ALS patients assessments. We have now clarified this information in the methods and result sections. For the comparison between ALS patient and identical healthy brother we performed 25 independent assessments from June 2018 to Nov of 2020. For other ALS patients compared to age and sex matched healthy donors we had 12 different healthy donors and 7 different ALS patients and we performed 37 different assessments for each group from June 2018 to Nov of 2020. The results were important since multiple assessments were done on several ALS patients to see whether disease progression corresponded to maintenance of increased CD8+ T cell function.

There are many additional issues with this manuscript. There is no clinical and genetic description of the patients and controls, the description of the methods is incomplete (particularly regarding the NAC treatment), it is unclear it the IRB approval also covered NAC injections,

many of the data presented are not statistically significant, it is unclear what the n represents in most experiments (number of technical replicates or biological replicates?)

Response: We have now included more clinical and genetic analysis and expanded on the description of methods on NAC treatment. We have indicated that NAC was administered as part of the patient care by the treating physician and we only received the schedule of treatments for the patient.

We have also clarified “n” in Material and Method section. Thank you

some of the data seem to be based on only one patient and have no standard deviations (only one experiment? e.g. Fig 7 and 8).

Response: These are representative data of minimum 5 experiments which we now clarified in each figure legend. Thank you

Reviewer 3 Report

This paper aims to describe differences in immune subsets in patients with ALS compared to healthy individuals. However, the used methodology is seemingly flawed. In figure 1, CD16 is used as a marker to determine NK cells from CD45+ PBMCs. This is not at all correct, as amongst other cells, CD8 T cells and monocytes also express CD16. If there was a marginal contamination with neutrophils, also neutrophils would be in the CD16+ population. As this is the premise of this paper, I don’t see any added value in continuing my review before the below major comments are addressed. As it is now, this manuscript cannot be considered for publication due to several major flaws.

Major comments

Line 116 – To my knowledge, CD16 is not exclusively an NK cell marker (neutrophils, CD8 T cells and and monocytes express CD16, for starters) and therefore not at all appropriate to ascertain purity. Please provide reference data that shows that CD16 is sufficient, for instance by comparing staining of CD56, CD57 and CD16 in on total material and after isolation. I’m fine if this is performed on healthy donor material.

Line 146 – 24 hours PMA/ionomycin stimulation is extremely toxic and not at all a suitable stimulation, especially at high concentrations. Especially if secreted proteins are investigated, as these cells will explode. Please provide rationale and control data that what is observed is not extreme cellular stress. Furthermore – line 166 – what should have been employed is chambers with cells that have not been stimulated. This is not at all an appropriate control. Please provide the control mentioned.

Line 210 – this should not at all result in Treg differentiation, but rather general CD4 T cell expansion.

Figure 1 – see my main comment 1 – CD16 on it’s own is not at all sufficient to gate out NK cells. This just says something about CD16-expressing cells. Also – this figure is completely overwhelming. Authors should try to make it more legible by either moving irrelevant data to the supplement, splitting up the graphs as applicable, or making 2 figures, one about subsets, one about killing, one about IFNg.

After this, I just scrolled through the manuscript – please correct the quality of figure 3, 6, 7 and 8, and make figure 5 readable, it’s not at all interpretable as it is now.

Minor comments

Line 45 – add comma after to date.

Line 46 – add comma after patients

Line 47 – swap is and therefore around

Line 49 – have also been found

Line 53 – finding singular, remove s

Line 54 – Monocyte and T cell singular, remove s 2x

Line 57 – Although more studies

Line 58 – the role of CD8+ T cells has

Line 59 – please provide reference(s) – this whole part needs to be more clearly referenced which sentence draws from what reference.

Line 66 – please provide the reference for the mouse study, and the reference for the gene implications.

Line 68 – add comma after reasons.

Line 69 – ALS patients

Line 73 – We previously reported that NAC treatment results in stem cell differentiation – also, which type(s) of stem cells? Please add.

Line 77 – is this about the current paper or the paper referenced in the previous sentence? Please clarify.

Line 82 – aged matched healthy donors?

Line 83 – please remove clearly.

Line 84 – relentless? Cells do not have feelings, please choose something more appropriate.

I am stopping here to provide input on the way the text is written, and will now focus on the content. I would strongly advise authors to seek help from a native English speaking colleague – which shouldn’t be too difficult seeing as they are based in California, or make use of a (professional) editing service, as just in the first couple of paragraphs, there are quite some things that need tweaking to fit better with scientific literature and adhere to basic syntax/grammar rules.

Line 98 – OSCSC? Please write out in full – this goes for all abbreviations -> I see this is now later in this paragraph, please move up.

Line 99 – please provide the clones for CD16 and CD3 and CD28, and the dilutions used.

Line 111 – described before = described previously, please correct throughout.

Line 114 - capitalize Stem Cell Technologies.

Line 131 – not at all complete – add all employed antibodies/cell stains, manufacturers, clones, fluorophores and dilutions to this manuscript for transparency.

Line 136 – different numbers? Please use effector to target cell ratios, or something of the sort, that is more commonly used to describe these experiments.

Line 171 – whats with all the numbers at the end of this sentence?

Author Response

August 19, 2022

We appreciate the hard work of our reviewers and have addressed their concerns. Below please find point by point response to the reviewers’ comments. We hope that the explanations and significantly modified version of the manuscript is now satisfactory for publication.

Thanks

Dr. Jewett

This paper aims to describe differences in immune subsets in patients with ALS compared to healthy individuals. However, the used methodology is seemingly flawed. In figure 1, CD16 is used as a marker to determine NK cells from CD45+ PBMCs. This is not at all correct, as amongst other cells, CD8 T cells and monocytes also express CD16. If there was a marginal contamination with neutrophils, also neutrophils would be in the CD16+ population. As this is the premise of this paper, I don’t see any added value in continuing my review before the below major comments are addressed. As it is now, this manuscript cannot be considered for publication due to several major flaws.

Response: We thank the reviewer and apologize for our oversight. The population of NK cells were assessed using CD3, CD16 and CD56. They were CD3-/CD16dim/+CD56dim/+. Therefore, we did not just use CD16 for the assessment.

Major comments

Line 116 – To my knowledge, CD16 is not exclusively an NK cell marker (neutrophils, CD8 T cells and and monocytes express CD16, for starters) and therefore not at all appropriate to ascertain purity. Please provide reference data that shows that CD16 is sufficient, for instance by comparing staining of CD56, CD57 and CD16 in on total material and after isolation. I’m fine if this is performed on healthy donor material.

Response: We thank the reviewer and apologize for our oversight. The population of NK cells were assessed using CD3, CD16 and CD56. They were CD3-/CD16dim/+CD56dim/+. Therefore, we did not just use CD16 for the assessment.

Our lab specializes in NK cell studies and have been studying these cells in the past 30 years and have published extensively on the phenotype and function of these cells. It was an oversight in our part, since our associates by thinking they may simplify the graph by just labeling with CD16 it will suffice, however, we have now corrected the figure and inserted the CD3-/CD16dim/+CD56dim/+ in the figure. We have now moved this figure to the supplement as suggested below and by other reviewers.

Line 146 – 24 hours PMA/ionomycin stimulation is extremely toxic and not at all a suitable stimulation, especially at high concentrations. Especi ally if secreted proteins are investigated, as these cells will explode. Please provide rationale and control data that what is observed is not extreme cellular stress. Furthermore – line 166 – what should have been employed is chambers with cells that have not been stimulated. This is not at all an appropriate control. Please provide the control mentioned.

Response: These experiments were performed by the company with their established protocol and criteria for testing. We only shipped the samples for on chip testing. They did not do 24-hour PMA/Iono stimulation for NK cells experiment. What they did was an on-chip PMA/Iono stimulation for NK cells, which was about 13 hours. For T cells, they always do 24-hour CD3/CD28 stimulation and then load the chip. Therefore, we apologize if the process was not clearly understood. The M&M about NK/CD8 T cells’ IsoCode experiment in the manuscript is “NK cells were first labeled with membrane stain (1:500 dilution, IsoPlexis) and then resuspended in complete RPMI medium at a density of 1 x 106 cells/mL with an addition of PMA (5 ng/ml; Sigma-Aldrich, P8139-1MG)) and Ionomycin (500 ng/ml; Sigma-Aldrich, 10634-1MG) for on chip stimulation. CD8+ T cells were treated with plate-bound anti-human CD3 (10 µg/ml; clone OKT3, Thermo Fisher/Invitrogen) and soluble anti-human CD28 (5 µg/mL; clone CD28.2, Thermo Fisher/Invitrogen) at a density of 1 x 106 cells/mL for 24 hours at 37°C, 5% CO2. Cells on the chip were incubated at 37°C, 5% CO2 for additional 13.5 hours on IsoLight automation system (IsoPlexis). We hope that this explanation is satisfactory to the reviewer. For the control, without stimulation, cells secrete none or very minimal cytokines according to company’s internal data. Therefore, in consultation with the head of company we decided to not include no stimulation samples since it would have significantly increased the cost of the experiment, in which the company even did not recommend or suggest to run since in thousands of samples that they ran previously they could not see meaningful increase in secretion under any disease condition.

Line 210 – this should not at all result in Treg differentiation, but rather general CD4 T cell expansion.

Response: We have now expanded on the description and added more text to the methodology. ImmunoCult™ Human Treg Differentiation Supplement Catalog # 10977 is a Serum-free culture supplement for the differentiation of human naïve CD4+ T cells into regulatory T cells (Tregs). This kit was purchased from stem cell technologies and used as suggested by the manufacturer to differentiate naïve CD4+ T cells to T reg cells. Based on manufacturers description ImmunoCult™ Human Treg Differentiation Supplement contains a human cytokine and small molecule formulated to promote the robust activation, expansion, and differentiation of peripheral blood-derived, naïve, CD4+ human T cells into regulatory T cells (Tregs). We just followed the suggestion from the manufacturer, and Stem Cell technologies is highly regarded company.  

Figure 1 – see my main comment 1 – CD16 on it’s own is not at all sufficient to gate out NK cells. This just says something about CD16-expressing cells.

Response: We thank the reviewer and apologize for our oversight. The population of NK cells were assessed using CD3, CD16 and CD56. They were CD3-/CD16dim/+CD56dim/+. Therefore, we did not just use CD16 for the assessment.

Also – this figure is completely overwhelming. Authors should try to make it more legible by either moving irrelevant data to the supplement, splitting up the graphs as applicable, or making 2 figures, one about subsets, one about killing, one about IFNg.

Response: We have now moved flow cytometry data to the supplementary file. Thank you

After this, I just scrolled through the manuscript – please correct the quality of figure 3, 6, 7 and 8, and make figure 5 readable, it’s not at all interpretable as it is now.

 Response: We have now improved the quality of figures. Thank you

Minor comments

Line 45 – add comma after to date.

Response: Done. Thank you

Line 46 – add comma after patients

Response: Done. Thank you

Line 47 – swap is and therefore around

Response: Done. Thank you

Line 49 – have also been found

Response: Done. Thank you

Line 53 – finding singular, remove s

Response: Done. Thank you

Line 54 – Monocyte and T cell singular, remove s 2x

Response: Done. Thank you

Line 57 – Although more studies

Response: Done. Thank you

Line 58 – the role of CD8+ T cells has

Response: Done. Thank you

Line 59 – please provide reference(s) – this whole part needs to be more clearly referenced which sentence draws from what reference.

Response: Done. Thank you

Line 66 – please provide the reference for the mouse study, and the reference for the gene implications.

Response: Done. Thank you

Line 68 – add comma after reasons.

Response: Done. Thank you

Line 69 – ALS patients

Response: Done. Thank you

Line 73 – We previously reported that NAC treatment results in stem cell differentiation – also, which type(s) of stem cells? Please add.

Response: Done. Thank you

Line 77 – is this about the current paper or the paper referenced in the previous sentence? Please clarify.

Response: Done. Thank you

Line 82 – aged matched healthy donors?

Response: Age and gender matched. Clarified in text. Thank you

Line 83 – please remove clearly.

Response: Done. Thank you

Line 84 – relentless? Cells do not have feelings, please choose something more appropriate.

Response: Done. Thank you

I am stopping here to provide input on the way the text is written, and will now focus on the content. I would strongly advise authors to seek help from a native English speaking colleague – which shouldn’t be too difficult seeing as they are based in California, or make use of a (professional) editing service, as just in the first couple of paragraphs, there are quite some things that need tweaking to fit better with scientific literature and adhere to basic syntax/grammar rules.

Response: We have now extensively changed the language and rewrote some of the paragraphs. We hope we have now addressed the reviewer’s concern adequately. Thank you

Line 98 – OSCSC? Please write out in full – this goes for all abbreviations -> I see this is now later in this paragraph, please move up.

Response: Done. Thank you

Line 99 – please provide the clones for CD16 and CD3 and CD28, and the dilutions used.

Response: Done. Thank you

Line 111 – described before = described previously, please correct throughout.

Response: Done. Thank you

Line 114 - capitalize Stem Cell Technologies.

Response: Done. Thank you

Line 131 – not at all complete – add all employed antibodies/cell stains, manufacturers, clones, fluorophores and dilutions to this manuscript for transparency.

Response: Done. Thank you

  Line 136 – different numbers? Please use effector to target cell ratios, or something of the sort, that is more commonly used to describe these experiments.

Response: Clarified in text. Thank you

Line 171 – whats with all the numbers at the end of this sentence?

Response: We apologize, these were repetition of references which we fixed now. Thank you

Round 2

Reviewer 2 Report

I appreciate the response of the authors to the previous concern regarding the number of the patients sampled. However, there are still many issues with the manuscript that were not addressed by this mostly cosmetic revision of the paper that greatly affect its quality.

First, the authors mention in their response that patients were sampled at multiple time points during disease progression "to see whether disease progression corresponded to maintenance of increased CD8+ T cell function". However the data are not presented as a longitudinal time series, but pooled together as a one time point. Second, many (most) of the data are still shown as "one of X representative experiments". This is not acceptable as the reader is not able to assess the reproducibility of the data and significance of the difference. The results from the multiple biological replicates MUST be pooled together and analyzed for statistical significance. Third, the authors have not improved much on their description of the patients enrolled in the study. A table summarizing the ages and genders of enrolled patients should be included. Importantly, the authors only mention in the discussion that the ALS twin had two ALS-causing mutations in two known genes, while its healthy sibling had one in a different gene. The authors also mention additional mutations but do not specify what genes. This information should be presented clearly in the methods section and in the result section. A much improved description of this data should be provided as it may be important to interpret the biological results.

Other concerns:

- In the introduction the authors mention 4 genes (C9ORF72, SOD1, TDP-43, FUS) as genetically linked to ALS. There are more than 25 genes associate with ALS to date. The authors should clarify their statement to reflect the complexity of ALS genetics.

- In the introductions (lines 78-79) the authors state that NAC anti-apoptotic activity is "through its ability to differentiate the stem cells". These two activities are linked but distinct as NAC antiapoptotic activity is not limited to stem cells.

- Results section (lines 269-270) "...percentages of CD14+ monocytes, CD4+ T, CD8+ T, and NKT cells in ALS patients’ PBMCs in comparison to healthy  individuals (Figs. S1a and S1b)." This data is not included in figure S1.

- Figure 2g is confusing as it is not clear what colors represent in the "polyfunctionality" and "Dominant functional group".

- Figure 6a: data should be quantified 

- Discussion: the statement in lines 589-594 claiming that the data presented support the idea of polygenic origin of ALS is a gross overinterpretation of the data

Author Response

August 29, 2022

We appreciate the hard work of our reviewers and have addressed their concerns. Below please find point by point response to the reviewers’ comments. We hope that the explanations and significantly modified version of the manuscript is now satisfactory for publication.

Thanks

Dr. Jewett

I appreciate the response of the authors to the previous concern regarding the number of the patients sampled. However, there are still many issues with the manuscript that were not addressed by this mostly cosmetic revision of the paper that greatly affect its quality.

First, the authors mention in their response that patients were sampled at multiple time points during disease progression "to see whether disease progression corresponded to maintenance of increased CD8+ T cell function". However the data are not presented as a longitudinal time series, but pooled together as a one time point.

Response: We have now added the data on the chronological basis too (Fig. 10). This has prompted us to expand the results and discussion significantly since we had to correlate observations based on the treatments received. This manuscript is already extensive and by adding more data we are just going to increase the size of the manuscript which we hope will not cause issues with the other reviewers and with the publishing company, since we still had to move data from the main manuscript to the supplementary file to fulfill their concerns. This was one reason we moved many important data to supplementary file, which we believe should otherwise be in the main manuscript, in order to decrease the size of the manuscript.

In addition, in this manuscript we had planned to present the important and key differences in immune function between the ALS patient and genetically identical twin brother globally, comparing their results to those patients who did not have genetically identical twin but were compared to those of healthy donors with no history of ALS. Furthermore, we had planned to focus in a follow up paper on the ALS patient and his twin brother in order to present the data in the chronological order and provide an in-depth discussion regarding the few timepoints that did not follow the trend of increase in the function of CD8+ T cells, which we have now done in this paper, albeit due to the constraint of the length of paper we had to minimize results and discussion. After all, these data are important and should be given proper space and attention to be presented since it is a very rare and precious opportunity that one can get to have a patient that have genetically identical twin which can be studied chronologically, and report the findings. Such mechanistic studies are of outmost significance for the advancements of effective treatments which are urgently needed in these patients.

Second, many (most) of the data are still shown as "one of X representative experiments". This is not acceptable as the reader is not able to assess the reproducibility of the data and significance of the difference. The results from the multiple biological replicates MUST be pooled together and analyzed for statistical significance.

Response: We are also showing the compiled data now. For certain assessments significant variability can be seen depending on the day of the experiment and the methodologies used. Inherent variabilities between different donors and internal variabilities that are within the methods used make the analysis in certain instances not representative of the single patient results.  Therefore, to avoid issues and present the patient data without confounding factors accounting for such variabilities we opted to demonstrate single representative experiment. However, we have now added the compiled results too. The reported data is from the multiple biological replicates performed at different dates from the patients and they are not experimental replicates.

Third, the authors have not improved much on their description of the patients enrolled in the study. A table summarizing the ages and genders of enrolled patients should be included.

Response: We have included a table for the patients in the supplemental section (Table S1). We have now limited to the patients that we had performed a number of assessments for each patient because it is important not only to see the trend with several assessments and not just once, and whether that trend remains high or it is an occasional occurrence is of high priority and therefore, important to report. 

 Importantly, the authors only mention in the discussion that the ALS twin had two ALS-causing mutations in two known genes, while its healthy sibling had one in a different gene. The authors also mention additional mutations but do not specify what genes. This information should be presented clearly in the methods section and in the result section. A much improved description of this data should be provided as it may be important to interpret the biological results.

Response: We have now presented this data in the result section and in the methods. The manuscript is already very large and including more data is going to only increase the size of the manuscript. We are constricted by how much data we can present in one paper. However, we have now added the genetic data to the manuscript. We hope that this is not going to cause issues with the other reviewers since they had suggested shortening some of the data and not increase in the numbers.

We have now added significant numbers of new data to results and methods and discussion to the already large paper, we assume that we will not be penalized and asked for shortening of the manuscript.

Other concerns:

- In the introduction the authors mention 4 genes (C9ORF72, SOD1, TDP-43, FUS) as genetically linked to ALS. There are more than 25 genes associate with ALS to date. The authors should clarify their statement to reflect the complexity of ALS genetics.

We did indeed mention in the revised version the fact that “appears to be heterogeneous. To date, over 20 genes have been found to be associated with ALS [2].”We have now expanded the sentence to indicate the complexity of ALS genetics “appears to be heterogeneous. To date, over 20 genes have been found to be associated with ALS which indicates the complexity of the ALS genetics [2]

- In the introductions (lines 78-79) the authors state that NAC anti-apoptotic activity is "through its ability to differentiate the stem cells". These two activities are linked but distinct as NAC antiapoptotic activity is not limited to stem cells.

We have now modified the sentence to read “N-acetyl cysteine (NAC) inhibits cell death due to its anti-oxidant activity, in part through its ability to differentiate the stem cells. It is also likely that NAC could protect differentiated cells from undergoing cell death.”

- Results section (lines 269-270) "...percentages of CD14+ monocytes, CD4+ T, CD8+ T, and NKT cells in ALS patients’ PBMCs in comparison to healthy  individuals (Figs. S1a and S1b)." This data is not included in figure S1.

Response: Please check revised supplementary file, this is Fig. S1.

- Figure 2g is confusing as it is not clear what colors represent in the "polyfunctionality" and "Dominant functional group".

We apologize for the non-clarity of colors. In the bottom left hand corner of the figure g there is a legend which indicates the colors and the respective population of cells with specific treatments, which are shown in the graphs. These parameters are set by the company and they are specialized in running these experiments, and run these for thousands of their clients.  

- Figure 6a: data should be quantified

We have now quantified the data and shown in the supplementary file since the manuscript has now many additional figures. Both the number of viable cells and the number of dead cells have also been shown in the Supplementary figure 7.

- Discussion: the statement in lines 589-594 claiming that the data presented support the idea of polygenic origin of ALS is a gross overinterpretation of the data

We did not claim the data presented supported the idea of polygenic origin of ALS, indeed we also presented the counter argument in the following sentences, that these are hypothesis and not facts, because they are not known at present. We stated “Although there are differences in the gene mutations between the patient and the healthy identical twin, they also shared a number of gene mutations (unpublished observation). Therefore, it is possible that other factors are involved in combination with the genetic abnormalities for the disease manifestation and progression. Alternatively, gene mutations in TARDBP and FUS may be sufficient and necessary for disease manifestations since these mutations were seen in many ALS patients. Of interest, is the observation of PRF-1 gene mutation in the ALS patient (unpublished observation), since perforin is highly upregulated in killer T cells and NK cells, and therefore, its mutation may play a role in the pathogenesis of the ALS disease.” 

Reviewer 3 Report

Dear authors, thank you for the clarification regarding the use of CD56 to identify NK cells, this was very reassuring. I would also like to thank you for the updates on the manuscript. Unfortunately, I cannot completely judge the contents yet due to some missing information and poor quality of some of the figures. I am happy to re-review once this has been amended. Please find my comments below regarding the improved manuscript.

Figure 1 – the chromium release data is still in this figure but not mentioned in the figure title, please ammend. Something like “Increased IFN-g secretion but not cytotoxicity by…” would do

Line 353 – please provide the reference here to this method as well, for ease of readers.

Line 443 – “When assessing IFN-g levels” does not fit with the statement after, please have a look at this sentence.

Line 449 – a heatmap does not exhibit anything, it’s a way to show when cells exhibit something. Please amend.

Figure 4 – I cant interpret all the panels as the figure does not fit on the page. Please amend.

Figure 5 – I can’t interpret the panels unless zooming in tremendously, as I cannot read what each panel represent. Please make the figure bigger by placing fewer analytes on one row, and making the figure longer (i.e. more horizontal). It’s fine if this is a page-fitting figure, if that’s what it takes for it to be legible.

Furthermore, I have performed many a Luminex experiment, and these types of measurements come with a lower limit of detection. Anything below that is, at best, hard to interpret. With the low levels measured in these sera, I would like to ask the authors to add the LOD for each analyte to the graph, i.e. by a hatched line. From the materials and methods, I can’t see which manufacturer was use as a supplier. If a kit from ThermoFisher was used, the LOD for IFN-g for instance is something around 9.5 pg/mL. If authors used at-home labelling, then of course no LODs are known, but either way, the used reagents need to be incorporated into the materials and methods (I searched for “Luminex” and checked the what I thought were relevant sections but couldn’t find information on this; if all the info is in Reference 29, that’s not going to be very helpful, this also needs to be added to this manuscript. I tried to check this reference myself but unfortunately my library does not subscribe to JI anymore).

Figure 8 – This figure is stretched wide in a way that it’s hard to read. Please amend.

Author Response

August 29, 2022

We appreciate the hard work of our reviewers and have addressed their concerns. Below please find point by point response to the reviewers’ comments. We hope that the explanations and significantly modified version of the manuscript is now satisfactory for publication.

Thanks

Dr. Jewett

Dear authors, thank you for the clarification regarding the use of CD56 to identify NK cells, this was very reassuring. I would also like to thank you for the updates on the manuscript. Unfortunately, I cannot completely judge the contents yet due to some missing information and poor quality of some of the figures. I am happy to re-review once this has been amended. Please find my comments below regarding the improved manuscript.

Figure 1 – the chromium release data is still in this figure but not mentioned in the figure title, please ammend. Something like “Increased IFN-g secretion but not cytotoxicity by…” would do

Response: We have now updated the title ” Significant increased IFN-γ secretion but not NK cell-mediated cytotoxicity was seen in PBMCs of ALS patients”

Line 353 – please provide the reference here to this method as well, for ease of readers.

Response: We have now added the reference.. osteoclast-mediated NK expansion methodology [39]

Line 443 – “When assessing IFN-g levels” does not fit with the statement after, please have a look at this sentence.

Response: We modified the sentence, now it reads as” Significantly higher granzyme B and MIP-1b secretion frequency in CD8+ T cells of ALS patients was seen as compared to those from the healthy controls”.

Line 449 – a heatmap does not exhibit anything, it’s a way to show when cells exhibit something. Please amend.

Response: We agree with the reviewer that heatmap is a quick visual tool to see differences on the global level. However, for these studies the heat map indicates how often a combination of proteins was observed in a given sample. Please see the description of the heatmap in the bottom of the figure. We have now changed the figure legend to reflect that.

Figure 4 – I cant interpret all the panels as the figure does not fit on the page. Please amend.

Response: We now have divided Fig. 4 in two figure, Fig. 4 and Fig. 5.

Figure 5 – I can’t interpret the panels unless zooming in tremendously, as I cannot read what each panel represent. Please make the figure bigger by placing fewer analytes on one row, and making the figure longer (i.e. more horizontal). It’s fine if this is a page-fitting figure, if that’s what it takes for it to be legible.

Response: We now have divided original Fig. 5 in two figure, Fig. 5a is now Fig. S6 and Fig. 5B is now Fig. 6.

Furthermore, I have performed many a Luminex experiment, and these types of measurements come with a lower limit of detection. Anything below that is, at best, hard to interpret. With the low levels measured in these sera, I would like to ask the authors to add the LOD for each analyte to the graph, i.e. by a hatched line. From the materials and methods, I can’t see which manufacturer was use as a supplier. If a kit from ThermoFisher was used, the LOD for IFN-g for instance is something around 9.5 pg/mL. If authors used at-home labelling, then of course no LODs are known, but either way, the used reagents need to be incorporated into the materials and methods (I searched for “Luminex” and checked the what I thought were relevant sections but couldn’t find information on this; if all the info is in Reference 29, that’s not going to be very helpful, this also needs to be added to this manuscript. I tried to check this reference myself but unfortunately my library does not subscribe to JI anymore).

We appreciate the comment from the reviewer and now have added a sentence in the method section to indicate the range of detection indicated in the company brochure. For the kit used from Millipore the sensitivity of detection in serum is indicated between 0.4-55.8 pg/ml. All of the detected cytokines and chemokines in serum of the patient and the healthy control is above 0.4 pg/ml.

Figure 8 – This figure is stretched wide in a way that it’s hard to read. Please amend.

Response: We have now fixed the figure. Hopefully now is easy to read.